# Bayesian modeling of HFC production pipeline suggests growth in unreported CFC by-product and feedstock production

Stephen Bourguet [1] ✉ & Megan Lickley [1,2]

Observationally-derived emissions of ozone depleting substances must be scrutinized to maintain the progress made by the Montreal Protocol in protecting the stratospheric ozone layer. Recent observations of three chlorofluorocarbons (CFCs), CFC-113, CFC-114, and CFC-115, suggest that emissions of these compounds have not decreased as expected given global reporting of their production. These emissions have been associated with hydrofluorocarbon (HFC) production, which can require CFCs as feedstocks or generate CFCs as by-products, yet emissions from these pathways have not been rigorously quantified. Here, we develop a Bayesian framework to jointly infer emissions of CFC-113, CFC-114, and CFC-115 during HFC-134a and HFC-125 production. We estimate that feedstock emissions from HFC-134a production accounted for 90% (82–94%) and 65% (47–77%) of CFC-113 and CFC-114 emissions, respectively, from 2015–2019, while by-product emissions during HFC-125 production accounted for 81% (68–92%) of CFC-115 emissions. Our results suggest that unreported feedstock production in low- to middle-income countries may explain the unexpected emissions of CFC-113 and CFC-114, although uncertainties within chemical manufacturing processes call for further investigation and industry transparency. This work motivates tightened feedstock regulations and adds a reduction in CFC emissions to the benefits of the HFC phasedowns scheduled by the Kigali Amendment.

When released into the atmosphere, chlorofluorocarbons (CFCs) contribute to stratospheric ozone loss while heating the earth's surface with global warming potentials thousands of times stronger than $CO_2$ on a centennial timescale[1]. Due to their ozone depletion potential (ODP), the production of CFCs for most uses is banned by the Montreal Protocol; accordingly, the atmospheric mixing ratios of the most abundant CFCs (e.g., CFC-11 and CFC-12) have declined in recent years, and there have been initial signs of ozone recovery[2–6]. However, the detection of unexpected sources of CFC emissions in recent years[7–9] has underscored the need to continually evaluate the consistency of reported values with atmospheric observations.

Ensuring compliance with the Montreal Protocol requires that unexpected emissions of controlled substances be carefully considered – which in turn requires a thorough assessment of emissions from permitted sources. For example, while CFC production for emissive uses has been banned globally since 2010, ongoing emissions of these gases from reservoirs produced prior to 2010, such as refrigerators and foams, continue to be a source of emissions[10]. The quantity of ozone-depleting substances (ODSs) stored in these banks and the rate at which they are released have been the focus of recent work[11–13]. In addition, there is an exemption for regulated CFCs to be produced and entirely used as feedstocks in the production of other compounds[14], such

[1]Earth Commons, Georgetown University, Washington, DC, USA. [2]Science, Technology, and International Affairs Program, Georgetown University, Washington, DC, USA. ✉e-mail: stephen.bourguet@georgetown.edu

as hydrofluorocarbons (HFCs), with the requirement that this production is reported to the Ozone Secretariat of the United Nations Environment Program (UNEP)[14–16]. However, if a controlled substance is not isolated and is instead produced and consumed as part of a multi-step process in the same integrated chemical manufacturing facility, then it is considered an intermediate and reporting of its production is not required[15,16]. Furthermore, controlled substances may be produced as unwanted by-products during the manufacturing of other compounds, but there is no reporting requirement for this production. Facilities are encouraged to maintain best practices to minimize by-product emissions[16], but certain production processes do yield substantial emissions of unwanted by-products[17], including the emission of CFC-115 during HFC-125 production[17].

In this work, we focus on recent atmospheric observations of CFC-113, CFC-114, and CFC-115 which indicate that there may have been sustained emissions of these compounds from 2004–2019[18–22]. (Unless otherwise noted, we refer to the sum of CFC isomers by the dominant isomer; i.e., CFC-113 refers to CFC-113 + CFC-113a. Implications of this are discussed in the "Methods".) While the ozone depletion and surface warming caused by the emissions of minor CFCs from 2010–2020 have been estimated to be minimal, continued growth in emissions could negate some of the progress made by the Montreal Protocol[22], prompting further evaluation of these observations. Previous work has suggested that emissions from banked reservoirs of CFC-113, CFC-114, and CFC-115 cannot explain observationally-derived values[13], and while there was no sign of a decrease in the emissions of these compounds from sources other than banks from 2004–2019 (Fig. 1A), the globally-aggregated feedstock production of CFC-113 and CFC-114 reported to the Ozone Secretariat decreased during this time (Fig. 1B; data from ref. 23). Thus, an unknown source of emissions may have prevented emissions from decreasing as one would expect based on reporting from 2004–2019.

Two possible explanations for these unexpected emissions are that either the chemical manufacturing pipeline that produces and consumes CFC feedstocks became increasingly leaky or that reporting lagged actual feedstock production during this time. Given that emissions can occur at any point during the production, distribution,

and consumption of feedstocks, we refer to all emissions from this pipeline as feedstock emissions. The Medical and Chemical Technical Options Committee's (MCTOC) 2022 Assessment Report estimated that improvements in emissions abatement technologies led to a decrease in feedstock emission rates from 4% (3–5%) in the 1980s to around 2.5% (0.9–4%) in the modern-day (not including emissions during transportation)[16], implying that under-reporting of feedstock production may be the more likely scenario. However, emission rates of feedstocks and by-products during the production of fluorinated greenhouse gases are highly uncertain and are thought to vary widely depending on factors specific to each manufacturing facility[24] – the 2019 Refinement to the 2006 IPCC Guidelines on National Greenhouse Gas Inventories suggested an emission factor of 4% with an uncertainty range of 0.1–20%[24] – so it is possible that global mean feedstock leakage rates have increased as older facilities have aged and new facilities have been built in regions with fewer regulations[25].

Regardless, these explanations both implicate the production of hydrofluorocarbons (HFCs), which are the main end-products of manufacturing processes associated with CFC-113, CFC-114, and CFC-115 emissions[15,16,26]. In particular, the estimated growth of combined production of the refrigerants HFC-134a and HFC-125 from around 200 Gg · y⁻¹ in 2004 to 500 Gg · y⁻¹ in 2019 (shown in Fig. 1C, D; data from ref. 25) has been associated with the concurrent rise in a suite of CFC emissions[18–20,22,27]. However, these emissions have not been studied jointly or at a process level.

As is summarized in Fig. 2 (adapted from ref. 23) and described further in the Supplementary Note, there are multiple pathways for the production of HFC-134a and HFC-125[26,28–30], and the conversion efficiencies between feedstocks, intermediates, and end-products depend on the specific catalysts and reaction environments used[26]. These pathways are distinguished here by their unsaturated feedstocks, which are sequentially fluorinated into CFCs and/or hydrofluorinated into HCFCs and, ultimately HFC end-products[26]. The allocation of production between these pathways is not publicly known, although it has been reported that the trichloroethylene (TCE) pathway, which may emit HCFC-133a but not CFCs, is more commonly used for HFC-134a production[31–33]. Recent reports on the atmospheric abundance of the intermediate compound HCFC-133a suggest that HFC-134a

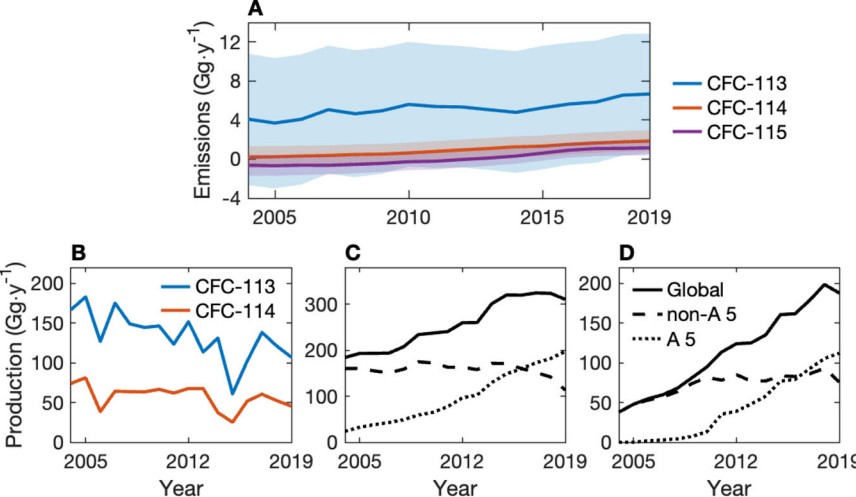

**Fig. 1 | Non-bank emissions of CFC-113, CFC-114, and CFC-115, and possible emission sources. A** The 5-y running mean of estimated observationally-derived global emissions of CFC-113 (blue), CFC-114 (orange), and CFC-115 (purple) that cannot be attributed to leakage from estimated banked reservoirs (i.e., non-bank emissions). **B** The globally aggregated production of CFC-113 (blue) and CFC-114 (orange) for use as a feedstock, as reported to the Ozone Secretariat (no CFC-115 production was reported during this time; data from ref. 23). **C** HFC-134a and (**D**)

HFC-125 estimated production globally (solid) and in Article 5 (dotted) and non-Article 5 countries (dashed) (data from ref. 25). In (**A**), the lines show median emissions, and the shaded regions encompass the 1-σ range of emissions based on the emission model and bank uncertainties. Global emissions are derived using the AGAGE 12-box model[46,63], and the calculation of bank emissions is described in the Methods. As the observational derived emissions time series end in 2020, the 3-y mean is used for 2019 in (**A**).

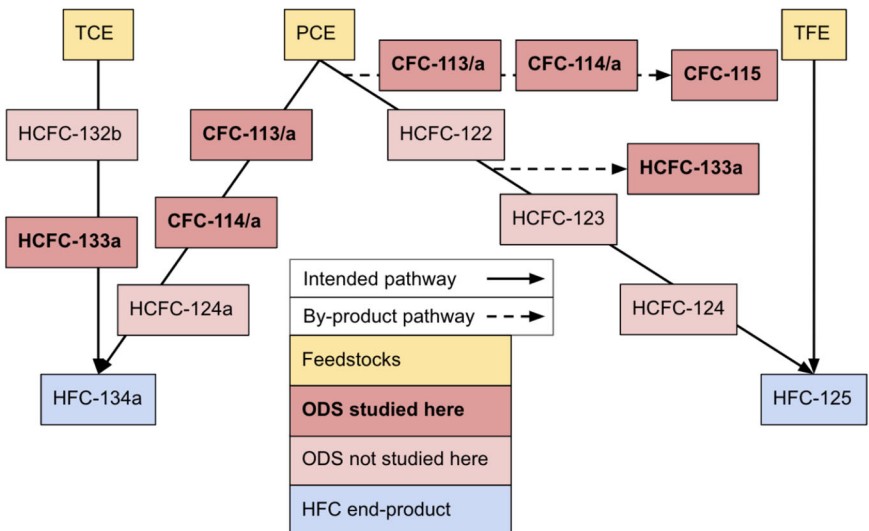

**Fig. 2 | A schematic of the known production processes for HFC-134a and HFC-125.** Production flowchart adapted from ref. 23, with feedstocks, intermediates, and by-products informed by relevant patents, as summarized in ref. 26. TCE = trichloroethylene, PCE = perchloroethylene, TFE = tetrafluoroethylene, ODS = ozone-depleting substance.

production by the TCE pathway may have increased since 2004, but the increase of observationally inferred HCFC-133a emissions has not been consistent, and it is possible that emissions may be influenced by facility-level containment practices[21,34]. For HFC-125, it was reported that 8 out of 12 production facilities in China used the tetrafluoroethylene (TFE) pathway[27], which is not known to emit CFCs as by-products, in 2011. However, it was also reported in 2023 that most HFC-125 was produced using PCE[17]. There is no known proxy for the TFE HFC-125 production pathway[29], so the usage of each production pathway cannot be inferred from observations. Given that the mass of each HFC produced by its corresponding production pathways determines feedstock, intermediate, and by-product generation, the unknown flow through the HFC production pipeline is a key source of uncertainty in attributing CFC emissions to HFC production.

To date, emission rates have been estimated for each gas in isolation by calculating the ratio of observationally inferred CFC emissions to HFC production[22,35]. This, however, does not account for the balance between production pathways in the chemical manufacturing pipeline or the efficiency of conversion between intermediate products. As such, the sources of emissions (i.e., feedstocks, banks, and by-products) have not been comprehensively quantified and reported feedstock and by-product emission rates[16,24] have not been constrained with atmospheric observations. Quantifying these emission rates could inform future controls of the Parties to the Montreal Protocol and add to the environmental benefits (i.e., reduced surface warming and ozone depletion) attributable to the Kigali Amendment[25], which is estimated to avoid 0.4 °C of warming by the end of the century through a phasedown of HFC production[7].

Here, we develop a probabilistic modeling approach, using Bayesian Parameter Estimation (BPE), that extends previous work[12] to jointly model CFC-113, CFC-114, and CFC-115 emissions from production, banks, feedstocks, and by-production as a function of their HFC-134a and HFC-125 end-products. We explicitly model HFC-134a production as the sum of two possible pathways (Fig. 2) and include observed HCFC-133a mixing ratios as an additional constraint on the relative production of HFC-134a through each pathway. (We chose HCFC-133a as a proxy for the TCE production pathway as HCFC-133a is the final intermediate of this pathway[26,28]. Relative to HCFC-132b, HCFC-133a is also preferable in that it has a lower lifetime uncertainty and thus simulated mixing ratios are better constrained[21].) We do not include other HCFC intermediates as constraints for HFC-125

production as HCFC-123 and HCFC-124 have other known end-uses[26], and HCFC-122 may be used as an intermediate in HCFC-123 and HCFC-124 production. Previously reported estimates of HFC-134a and HFC-125 production in Article 5 (A 5; low- to middle-income) and non-Article 5 (non-A 5; high-income) countries[25] are used to jointly model and constrain feedstock production and feedstock and by-product emission rates from the manufacturing pipeline in the two classifications of countries. We draw on previously reported emission rates[16,24], and production patents[26,28,36,37] are used to inform conversion rates between feedstocks, intermediates, and their HFC end products. By explicitly modeling the conversion and by-production of these CFCs and HCFC-133a through the HFC-125 and HFC-134a manufacturing pipelines in A 5 and non-A 5 countries, we attempt to explain the apparent discrepancy between trends in reported feedstock production and observationally derived emissions and quantify feedstock and by-product emission rates in each country classification. Finally, we provide a lower-bound estimate of the unintended $CO_2$-equivalent and CFC-11-equivalent emissions for HFC-125 and HFC-134a production, which quantifies the projected climate and ozone impact of their continued production under the Kigali Amendment.

## Results and discussion
### Simulated mixing ratios and emissions
The BPE posterior distributions of simulated CFC-113, CFC-114, CFC-115, and HCFC-133a surface mixing ratios accounting for emissions attributable to HFC production contain observations from 2004–2020 (Fig. 3, left column), confirming that our simulation model and parameter space are statistically consistent with observations. To quantify the impact of HFC production on observed mixing ratios, we compare BPE posterior mixing ratios with a scenario in which HFC-production-related emissions had not occurred from 2004–2019 (i.e., simulating mixing ratios using posterior emissions from non-HFC sources only from 2004 onwards). These results suggest that HFC production elevated the mixing ratios of CFC-113, CFC-114, and CFC-115 by 3.0 ppt (2.2–3.8 ppt), 0.9 ppt (0.6–1.2 ppt), and 0.6 ppt (0.4–0.7 ppt), respectively, in 2019. We assume that HCFC-133a is only emitted during HFC-134a and HFC-125 production; therefore, due to its relatively short atmospheric lifetime, we estimate that the mixing ratio of HCFC-133a would have decayed to less than 0.01 ppt in 2020 had there been no HFC production throughout this time period.

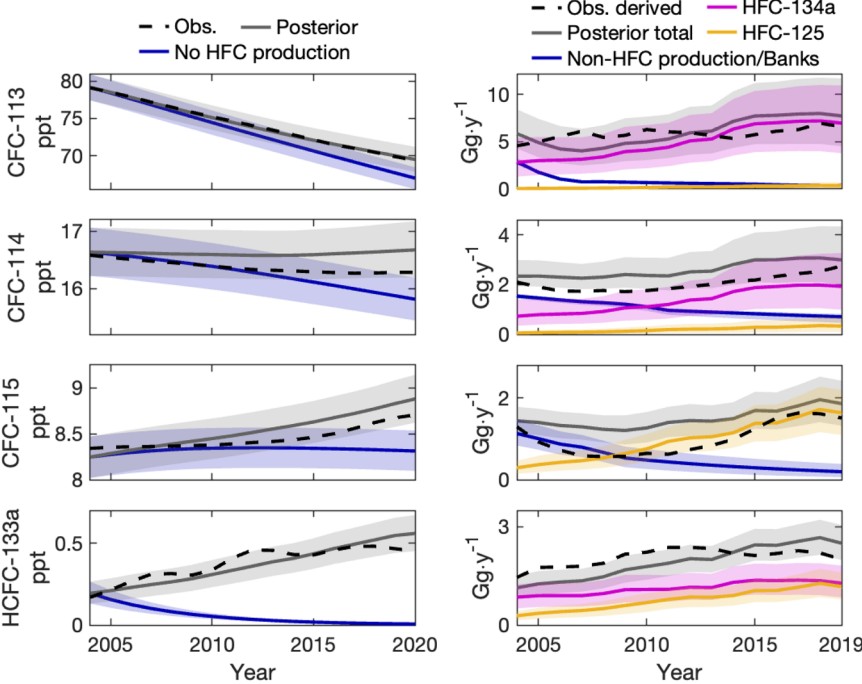

**Fig. 3 | Observed and simulated mixing ratios and emissions of CFC-113, CFC-114, CFC-115, and HCFC-133a.** (Left column) BPE posterior distributions of simulated global mean surface mixing ratios (gray), and simulated mixing ratio that use posterior emissions from non-HFC sources only (blue) for CFC-113 (top row), CFC-114 (second row), CFC-115 (third row), and HCFC-133a (bottom row). (Right column) BPE posterior distributions of total emissions (gray), and emissions attributable to non-HFC production and banks (blue), HFC-134a (magenta), and HFC-125 (gold). The dashed black lines are global mean surface observations and 5-y running mean observationally-derived emissions from AGAGE (3-y mean is shown for 2019), and the colored lines and shaded regions are the median and 1-$\sigma$ CI of each time series.

Relative to previous work[13], the magnitudes and trends of the BPE posterior emission distributions for these gases provide an improved comparison with observationally-derived emissions from 2004–2019 (Fig. 3, right column). As the central aim of this work is to capture the underlying multi-year trend in emissions, rather than year-to-year variability, the 5-y running mean of observationally-derived emissions is shown. Despite this smoothing, some variability remains that is not captured by our simulation posteriors, particularly for CFC-113 and HCFC-133a. It is possible that facility-level drivers of interannual emissions variability, such as leakage during maintenance or improvements to containment following modernization[34], could be a source of this variability – the BPE model assumes constant emission rates from all sources, and thus cannot capture these effects. The model also uses an HFC production time series that is partly informed by a top-down emission estimate, which would not capture potential temporal misalignment in production and emissions[25], although we anticipate that 5-y averaging should smooth out this variability. CFC-113 may also have been used to a lesser extent during this time as a feedstock or intermediate in the production of other end-products, such as chlorotrifluoroethylene (CTFE) plastics, trifluoroacetic acid (TFA), and the hydrofluoroolefin HFO-1336mzz(Z)[15,38,39]. Thus, the remaining variability in observationally-derived emissions could be from several sources that do not impact the overall trend of emissions associated with HFC production.

The estimated contribution of each emission source is also shown in the right column of Fig. 3. (As CFC production for most end-uses was phased out by 2010, we combine emissions from production for non-feedstock use and banks in these plots.) According to the BPE posterior distributions, bank emissions for CFC-113, CFC-114, and CFC-115 were approaching zero by 2019, while HFC production drove the overall increases in total emissions, consistent with previous work associating HFC production to CFC and HCFC emissions[19–22]. For CFC-113, our results attribute 81% (1-$\sigma$: 70–87%) of the sum of BPE estimated total emissions from 2004 to 2019 to HFC-134a production, including 90%

(82–94%) from 2015 to 2019. In contrast, we estimate that HFC-134a production did not contribute a majority of annual CFC-114 emissions until 2012, although it did account for 65% (47–77%) of emissions from 2015 to 2019. Our results also suggest that bank emissions were the largest source of CFC-114 prior to 2012, and they contributed 25% (17–36%) of emissions from 2015 to 2019. Meanwhile, we estimate that bank emissions were the dominant source of CFC-115 emissions through 2009, after which by-product emissions from HFC-125 production dominated, including contributing 81% (68–92%) of emissions from 2015 to 2019. Finally, we estimate that 59% (55–69%) of HCFC-133a emissions from 2004 to 2019 were the result of HFC-134a production, with the remaining portion attributable to HFC-125 production, although these sources were statistically equivalent by 2019.

### Emissions from HFC-134a production

As discussed above, HFC-134a can be produced via two different pathways, one of which consumes CFC-113 and CFC-114, while the other consumes HCFC-133a. Consistent with previous reports of the dominance of the TCE production pathway[31–33], which uses HCFC-133a, our BPE analysis suggests that this pathway accounted for 74% (60–85%) of global HFC-134a production from 2004 to 2019 (Fig. 4A). This percentage may have dropped as HFC-134a production grew in A 5 countries while decreasing in non-A 5 countries: We estimate that 66% (43–88%) of HFC-134a was produced via HCFC-133a in A 5 countries and 79% (58–94%) was produced via HCFC-133a in non-A 5 countries from 2004 to 2019.

As is shown in Fig. 4B, C, BPE estimated CFC-113 feedstock production grew from 83 Gg · y$^{-1}$ (40–151 Gg · y$^{-1}$) in 2004 to 205 Gg · y$^{-1}$ (116–290 Gg · y$^{-1}$) in 2019, while CFC-114 feedstock production grew from 83 Gg · y$^{-1}$ (36–158 Gg · y$^{-1}$) to 192 Gg · y$^{-1}$ (106–272 Gg · y$^{-1}$), and HCFC-133a production grew from 178 Gg · y$^{-1}$ (137–203 Gg · y$^{-1}$) to 261 Gg · y$^{-1}$ (211–313 Gg · y$^{-1}$). The estimated growth in the production of these compounds followed the growth in the production of HFC-134a by the pathway relevant to these compounds.

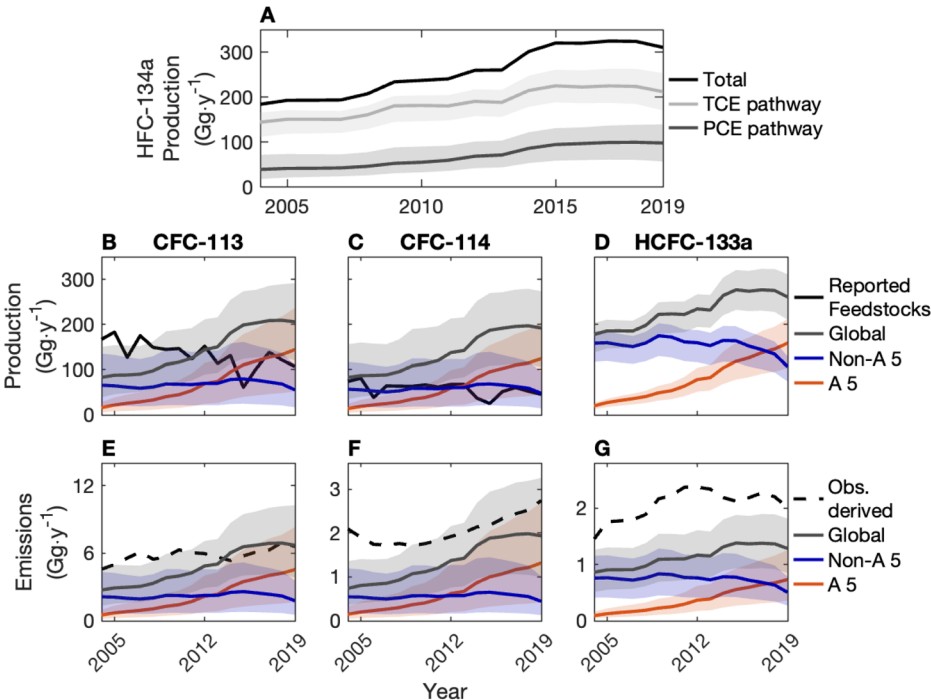

**Fig. 4 | Estimated HFC-134a feedstock and intermediate production and emissions. A** Estimated global HFC-134a production (black; data from ref. 25) and the BPE estimated mass of HFC-134a produced using tricholoroethylene (TCE) (light gray) and percholoroethylene (PCE) (dark gray) as feedstocks. **B–D** BPE posterior distributions of production and (**E–G**) emissions of CFC-113 (left column), CFC-114 (middle column), and HCFC-133a (right column). The lines and shaded regions are the median and 1-$\sigma$ CI, respectively, and in (**B–G**) the gray, blue, and orange coloring denotes global, non-Article 5, and Article 5 countries, respectively. For reference, the mass of feedstock production reported to the Ozone Secretariat[23] is included in black in (**B, C**), and the observationally derived emissions are included in dashed black in (**E–G**).

To assess the efficacy of current reporting practices, we compare the globally aggregated CFC-113 and CFC-114 feedstock production data reported to the Ozone Secretariat with our BPE estimated feedstock production (Fig. 4B, C). Notably, reported CFC-113 production values are greater than our 1-$\sigma$ interval for estimated non-A 5 production from 2004 to 2012 and at the high end of the interval from 2013 to 2019. Assuming that our HFC-134a production estimates in A 5 and non-A 5 countries are accurate, and given that reported values for CFC-113 came only from non-A 5 countries from 2008 to 2019[15], this suggests that feedstock production in non-A 5 countries was likely, not underreported during this time. As CFC-113 is known to be used as a feedstock in other emerging production processes[15,38,39], the low bias of the BPE posterior may reflect reporting of CFC-113 for use in these processes and accurate reporting of CFC-113 feedstock use across the industry. It is also possible that the assumed chemical conversion rate for the production of CFC-114 from CFC-113 is lower than our assumed value of 98%[26,36], which would increase the mass of CFC-113 feedstock required to produce a given mass of HFC-134a (see Supplementary Fig. 1). Nonetheless, our results suggest that there is a large and growing portion of CFC-113 and CFC-114 feedstock production going unreported, likely in A 5 countries. Following from the previously reported estimate of HFC production in A 5 and non-A 5 countries used to inform our priors[25], 63% (32–87%) of CFC-113 and CFC-114 production occurred in A 5 countries from 2015 to 2019, up from 27% (9–59%) in 2004–2008, reflecting a potential increase in the mass of global feedstock production that was not reported.

BPE posterior distributions of the global emission rates of CFC-113, CFC-114, and HCFC-133a relative to inferred feedstock production are provided in Table 1. The emission rate distribution is highest for CFC-113 – 3.4% (2.5–4.4%) globally from 2015–2019 – while CFC-114 and HCFC-133a feedstock emission rates were 1.0% (0.5–2.1%) and 0.5% (0.3–0.7%), respectively. This CFC-113 emission rate estimate is within the MCTOC likely range of 1.5–6.1%, suggesting that manufacturing facilities are operating as expected for well-regulated modern facilities[16]. However, it is possible that some quantity of emissions from other sources may be misattributed to HFC-134a production here, meaning that the emission rate for CFC-113 feedstocks used in HFC-134a production could be lower than our BPE posterior estimate. This is due to the omission of other feedstock uses for CFC-113 in this analysis, as well as the fact that our CFC-113 feedstock production estimate has a low bias relative to reported CFC-113 feedstock production. Meanwhile, the CFC-114 and HCFC-133a emission rate estimates are at the low end or below the MCTOC range[16] – one explanation for this could be that these compounds were not transported between production and consumption and could, therefore, be considered intermediates without reporting requirements. For HCFC-133a, this is consistent with previous reports of it being a non-isolated intermediate in the production of HFC-134a[40].

Table 1 also provides estimated feedstock emission rates of CFC-113, CFC-114, and HCFC-133a in A 5 countries and non-A 5 countries. Contrary to previous assumptions that A 5 countries emit at a higher rate[41], the estimated non-A 5 and A 5 feedstock emission rates are not statistically different at the 1-$\sigma$ confidence level. If our modeling assumptions are correct, this suggests that containment technologies are comparable across both sets of countries. As a result, the global feedstock emission rates have not changed as production has shifted to A 5 countries, and emissions in Fig. 4E–G have followed the same trends as inferred production.

Due to limited chemical conversion rates and the mass ratio between HFC-134a and its feedstocks, emission rates relative to HFC-134a production are higher than those relative to feedstock production itself. According to relevant patents and reports on the conversion processes[36,37], ~98% of CFC-113 can be converted into CFC-114a, and 94% of CFC-114a can be converted into HFC-134a. Meanwhile, the

**Table 1 | The median and 1-σ confidence intervals for BPE posterior distributions of feedstock emission rates (*FE* in Eq. 2) for species used in the production of HFC-134a**

| Species | Global | non-A 5 | A 5 |
|---|---|---|---|
| CFC-113 | 3.4% (2.5–4.4%) | 3.4% (2.2–4.7%) | 3.5% (2.2–4.8%) |
| CFC-114 | 1.0% (0.5–2.1%) | 1.1% (0.5–1.9%) | 1.3% (0.6–2.0%) |
| HCFC-133a | 0.5% (0.3–0.7%) | 0.5% (0.3–0.8%) | 0.5% (0.2–0.8%) |

Values are relative to the inferred mass of feedstocks produced (see Fig. 4B–D). Non-A 5 and A 5 emission rates do not vary with time, while the global emission rates may vary as production shifts from non-A 5 to A 5 countries. As such, the global values listed here are for the time mean of each percentile in the years 2015–2019.

**Table 2 | The median and 1-σ confidence intervals for BPE posterior distributions of by-product emission rates (BP in Eq. 2) for species emitted during the production of HFC-125**

| Species | Global | non-A 5 | A 5 |
|---|---|---|---|
| CFC-113 | 0.2% (< 0.1–0.3%) | 0.2% (< 0.1–0.4%) | 0.2% (< 0.1–0.4%) |
| CFC-114 | 0.2% (< 0.1–0.3%) | 0.1% (< 0.1–0.3%) | 0.2% (< 0.1– 0.4%) |
| CFC-115 | 0.7% (0.5–1.0%) | 0.8% (0.4–1.2%) | 0.9% (0.5–1.4%) |
| HCFC-133a | 0.7% (0.4–0.9%) | 0.8% (0.4–1.1%) | 0.5% (0.3–0.9%) |

Values are relative to the mass of HFC-125 produced (see Fig. 1D); percentages are therefore wt%. Non-A 5 and A 5 emission rates do not vary with time, while the global emission rates may vary as production shifts from non-A 5 to A 5 countries. As such, the global values listed here are for the time mean of each percentile in the years 2015–2019.

respective molar masses of CFC-113, CFC-114a, and HFC-134a are 187, 171, and 102 g mol$^{-1}$ – therefore, about 2 g of CFC-113 could be needed to produce 1 g of HFC-134a. By dividing BPE posterior emissions of CFC-113 and CFC-114 from HFC-134a production by the estimated mass of HFC-134a produced using the PCE pathway, our results suggest that the CFC-113 and CFC-114 emission rates relative to HFC-134a production from 2015 to 2019 were 7.2 wt% (5.2–9.4 wt%) and 2.1 wt% (1.3–3.2wt%), respectively. An analogous calculation for the HCFC-133a emission rate relative to the estimated mass of HFC-134a produced by the TCE pathway suggests an emission rate of 0.6 wt% (0.4–0.9 wt%).

**Emissions from HFC-125 production**
Given limited knowledge of by-product production, release, and destruction rates, it is not possible to determine the mass of by-products generated during HFC-125 production, so emission rates are reported in Table 2 relative to the mass of HFC-125 produced. Globally, the CFC-115 BPE estimated by-product emission rate from HFC-125 production was 0.7 wt% (0.5–1.0 wt%) from 2015 to 2019, which is consistent with the estimated range of 0.1–1 wt% recently reported by the UNEP's Technology and Economic Assessment Panel for this emission rate[17]. Following our modeling assumptions regarding the relative magnitude of CFC-115 emissions (see "Methods"[17,30]), the BPE estimated CFC-113 and CFC-114 emission rates from HFC-125 production (0.2 wt% (< 0.1–0.3 wt%)) were lower than that of CFC-115. These rates are higher than what was recently reported based on plant data (< 0.0001 wt%,[17]) but lower than the default emission factor of 4 wt% suggested by the 2019 Refinement to the 2006 IPCC Guidelines on National Greenhouse Gas Inventories[24].

Although the BPE estimated CFC-115 by-product emission rate from HFC-125 production is not inconsistent with UNEP's recent emission rate estimate, we expect our result to be biased low. As discussed above, it is not known how much HFC-125 is produced using the PCE pathway, which produces CFC-115 as a by-product, but it has been reported that only 4 out of 12 Chinese factories that produced HFC-125 in 2011 used this production pathway[27]. If global HFC-125 production follows the same ratio as Chinese factories, then the estimated CFC-115 emission rate from HFC-125 production would be 2–3 wt%, which would be closer to the 2019 Refinement to the 2006 IPCC Guidelines on National Greenhouse Gas Inventories value[24]. The BPE posterior emission rate of HCFC-133a from HFC-125 production, which does not have a specific previously estimated by-product emission rate, would also be within this 2–3 wt% range.

Table 2 shows that the BPE posterior HFC-125 by-product emission rate distributions are not higher in A 5 countries than in non-A 5 countries. Yet, despite the similarities in these rates, our results suggest that the rise in emissions from HFC-125 production from 2010 to 2019 was driven by an increase in production in A 5 countries. As is shown in Fig. 5, BPE estimated by-product emissions of CFC-115 and HCFC-133a from HFC-125 production in non-A 5 countries were flat during this time period, while emissions in A 5 countries followed the growth of HFC-125 production. Production of HFC-125 is expected to

be dominated by A 5 countries in the coming decades[25], so improved technology for the separation and containment of unwanted by-products during the production of HFC-125 in A 5 countries may be needed to prevent future emissions of CFC-113, CFC-114, and in particular, CFC-115.

**Ozone depletion and global warming potentials**
While HFCs do not destroy ozone, the CFC and HCFC emissions considered in this analysis will do so. Per Gg of HFC-134a and HFC-125 produced, our results suggest that unintended (i.e., feedstock and by-product) ODS emissions had an ODP weighting of about 0.021 ODP-Gg (0.013–0.029 ODP-Gg) and 0.006 ODP-Gg (0.004–0.008 ODP-Gg), respectively, from 2015 to 2019. Note that the lifetimes of CFC-113, CFC-114, and CFC-115 are longer than that of CFC-11, which ODP is calculated in reference to, so these numbers do not reflect the timing of ozone depletion. In particular, CFC-115 decays with a lifetime of 540 years[42], meaning that its impact on ozone depletion will be small in the near term and largely after the expected return to 1980 stratospheric chlorine and bromine levels.

Following from the increase in HFC-134a production, we estimate that the unintended ODP-weighted emissions from HFC-134a production grew from 2.0 ODP-Gg · y$^{-1}$ (1.0–3.4 ODP-Gg · y$^{-1}$) in 2004 to 4.9 ODP-Gg · y$^{-1}$ (2.9–7.2 ODP-Gg · y$^{-1}$) in 2019 (Fig. 6A). This is consistent with the combined ODP-weighted emissions of CFC-113 and CFC-114 feedstock emissions reported in Chapter 7 of the 2022 Scientific Assessment of Ozone Depletion (2.3–4.6 ODP-Gg · y$^{-1}$)[23]. Emissions of CFC-113, which has both the highest ODP of the ODSs considered here and the highest BPE estimated emission rate from HFC-134a production, account for 79% (71–87%) of the unintended HFC-134a ODP-weighted emissions over this time period. For HFC-125, the unintended ODP-weighted emissions are smaller, with a maximum of 1.2 ODP-Gg · y$^{-1}$ (0.9–1.5 ODP-Gg · y$^{-1}$) in 2018 (Fig. 6B). By gas, we estimate that CFC-115 contributed 60% (43–75%) of HFC-125's unintended ODP-weighted emissions from 2004 to 2019, while CFC-113 and CFC-114 contributed 19% (6–34%) and 15% (5–29%), respectfully. Emissions of HCFC-133a, which has a much lower ODP than any CFC, account for 0.8% (0.4–1.6%) and 3% (2–4%) of the ODP-weighted emissions for HFC-134a and HFC-125, respectively. We estimate that the total ODP-weighted emissions attributed to HFC-134a and HFC-125 production was 5.9 ODP-Gg · y$^{-1}$ (4.0–8.0 ODP-Gg · y$^{-1}$) from 2015 to 2019, which is about 7% of CFC-11 emissions during that time period[43].

By including the 100-year global warming potential (GWP) of unintended emissions, the total $CO_2$-equivalent of emissions attributable to HFC-134a and HFC-125 from 2004 to 2019 increased by 16% (11–20%) and 8% (6–11%), respectively (Fig. 6C–D). CFC-113 emissions from HFC-134a production had the largest $CO_2$-equivalent, which was 36 TgCO$_2$eq · y$^{-1}$ (21–56 TgCO$_2$eq · y$^{-1}$) in 2019, while the 16 TgCO$_2$eq · y$^{-1}$ (11–21 TgCO$_2$eq · y$^{-1}$) of emissions of CFC-115 was the largest $CO_2$-equivalent of the HFC-125 by-products in 2019. The $CO_2$-equivalent of combined feedstock and by-product emissions from

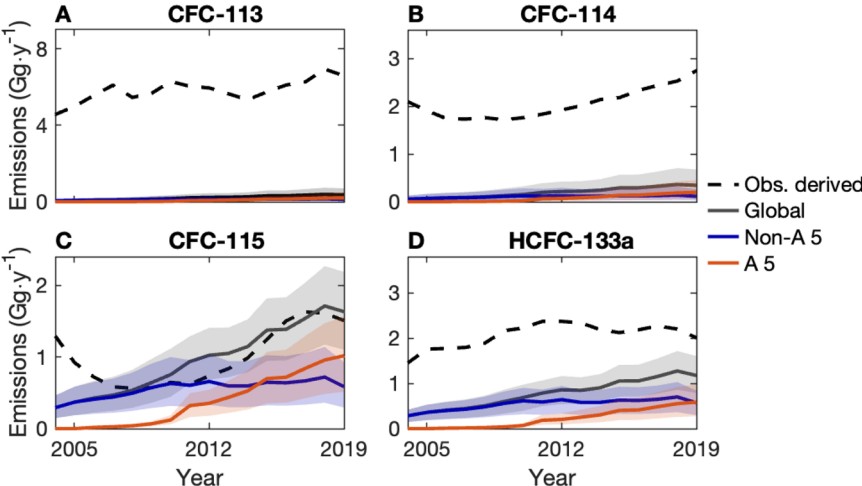

**Fig. 5 | Estimated HFC-125 by-product emissions. A** CFC-113, (**B**) CFC-114, (**C**) CFC-115, and (**D**) HCFC-133a BPE posterior distributions of by-product emissions from the manufacture of HFC-125. Gray lines denote global emissions, while blue and orange lines denote emissions in non-Article 5 and Article 5 countries, respectively. Lines and shaded regions are the median and 1-$\sigma$ CI, respectively, and observationally derived emissions are included as dashed black lines for reference.

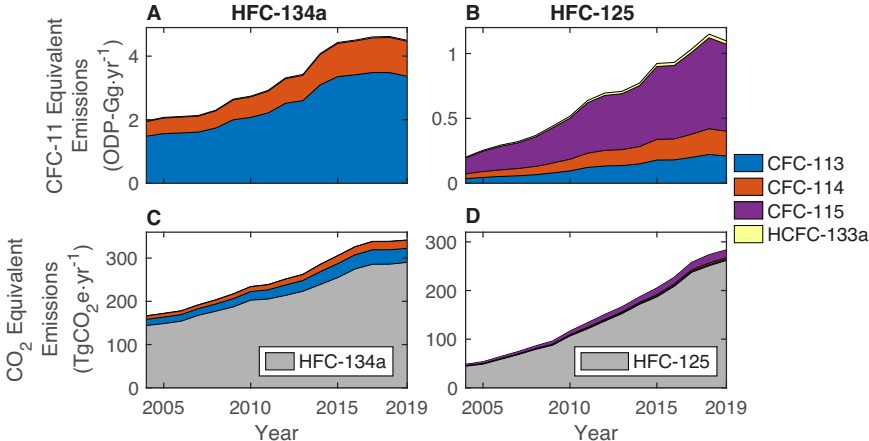

**Fig. 6 | The estimated environmental impact of feedstock, intermediate, and by-product emissions. A**, **B** The ozone depletion potential (ODP) and (**C**, **D**) global warming potential (GWP) of emissions attributed to (**A**, **C**) HFC-134a and (**B**, **D**) HFC-125 production, with the blue, orange, purple, and yellow sectors representing the contributions of CFC-113, CFC-114, CFC-115, and HCFC-133a, respectively. HFC-134a and HFC-125 have no ODP and are therefore not included in (**A**, **B**), while the GWP of HFC-134a and HFC-125 emissions are included for reference in gray in (**C**, **D**). GWP and ODP values were calculated with median emissions values; uncertainty ranges are presented in the text.

HFC-134a and HFC-125 production was 76 $TgCO_2$-eq · $y^{-1}$ (54–99 $TgCO_2$-eq · $y^{-1}$) from 2015 to 2019 – which is about 0.2% of the approximately 36,000 Tg · $y^{-1}$ of global $CO_2$ emissions during this time[44], increasing the total GWP attributable to HFC-134a and HFC-125 production to around 1.6% of global $CO_2$ emissions.

The ODP-weighted and CO2-equivalent emissions presented here do not include the impacts of other feedstocks, intermediates, or by-products that are released during the production of HFC-134a and HFC-125. Two such compounds, HCFC-31 and HCFC-132b, have been detected in the atmosphere in small abundances (less than 0.2 ppt) leading to emission estimates of about 1 Gg · $y^{-1}$ of each[21]. These compounds have ODPs of 0.019 and 0.038, respectively, and GWPs of 85 and 332, respectively; therefore, the total ODP and GWP of emissions related to HFC-134a and HFC-125 production is higher by about 1% due to these HCFCs. A full life-cycle analysis of HFC-134a and HFC-125 is outside of the scope of this work, but the contribution of $CCl_4$, which is a feedstock for PCE production, would need to be considered to capture the full ODP and GWP of these HFCs.

## Montreal Protocol reporting practices fall short in capturing increased CFC production

By jointly modeling the emissions of CFC-113, CFC-114, CFC-115, and HCFC-133a from reported non-feedstock production and from HFC-134a and HFC-125 production, we find that the increase in non-bank emissions of CFC-113, CFC-114, and CFC-115 from 2004 to 2019 can be explained by the concurrent increase in HFC production. Following our assumptions, we find that the use of CFC-113 and CFC-114 as feedstocks or intermediates during the manufacture of HFC-134a and the undesirable production of CFC-115 in a side reaction during the production of HFC-125 were likely the dominant sources of emissions for these compounds, respectively. In addition, our results suggest that HCFC-133a emissions during this time came primarily from its use as an intermediate in the production of HFC-134a, although its undesirable by-production during the manufacturing of HFC-125 may have contributed to the increase in emissions observed from 2004 to 2019.

Our results suggest that recent reporting of feedstock production is not sufficient to account for the production and emission of CFC-113, CFC-114, and CFC-115. From 2004 to 2019, reported production of CFC-

113 and CFC-114 for use as feedstocks decreased, while CFC-115 feedstock production was not reported. Notably, no feedstock production was reported in A 5 countries from 2008 to 2019[15]. As our modeled estimate of CFC-114 feedstock production in non-A 5 countries roughly aligns with reported values, while our CFC-113 non-A 5 feedstock production estimate is biased low relative to reported values, we assume that unreported production in non-A 5 countries is not a likely source of CFC-113 or CFC-114 emissions. Meanwhile, if our modeling assumptions are correct, we estimate that CFC-113 and CFC-114 were produced in increasing quantities for use as feedstocks in HFC-134a production in A 5 countries from 2004 to 2019. Thus, emissions from unreported feedstock production in A 5 countries may explain the discrepancy between trends in reported feedstock production and observationally-derived emissions during this time.

It has previously been suggested that the non-reporting of CFC-113 production in A 5 countries indicates its use as an intermediate, rather than a feedstock, in the production of other fluorinated compounds[15]. This distinction has practical implications for emissions – intermediates should be emitted at a lower rate – and regulatory implications for whether or not production is required to be reported under the Montreal Protocol. Due to the magnitude of uncertainty in our results relative to the precision needed to differentiate between use as a feedstock and intermediate, we cannot definitively say whether CFC-113 and CFC-114 were produced and consumed as intermediates or feedstocks. However, if the HFC-134a production estimate used here is accurate and non-A 5 countries produced and consumed feedstocks in separate processes, results from our analysis would suggest that A 5 countries have either not fulfilled their reporting obligations or have facilities that emit at a higher rate than non-A 5 facilities. The latter assumption is supported by the fact that CFC-113 and CFC-114 feedstock production was reported in non-A 5 countries from 2004 to 2019 – reporting would not have been required if production and consumption occurred in the same integrated process. (As our CFC-114 estimated emission rates are at the low end of what is thought to be technically feasible, it is plausible that some amount of CFC-114 feedstock production was produced and consumed as an intermediate.) If all steps in the production process emit at the same rate across the globe, then emissions from processing and transport are required for A 5 inferred emission rates to match non-A 5 inferred emission rates. Conversely, if CFC-113 and CFC-114 are produced and consumed as part of an integrated production process in A 5 countries, then results suggest some other part of the production process must emit at a higher rate than non-A 5 countries to compensate for the 0.3–1.2% emission rate that occurs during transportation[16]. If our modeling assumptions are correct, then A 5 countries have either produced unreported feedstocks or need to improve emission containment to match that of non-A 5 countries, consistent with a recent study of facility-level emission rates in China[45]. However, we emphasize that this conclusion is contingent on our assumptions regarding HFC production totals, the distribution of production between A 5 and non-A 5 countries, reporting practices in non-A 5 countries, chemical conversion rates, and the exclusion of additional emission sources, which represent critical sources of uncertainty that cannot be resolved in the present modeling framework.

**Lingering uncertainty**
The assumptions underlying our simulation of CFC emissions from HFC production are informed by published patents and estimated HFC production data. Nonetheless, biases in these assumptions would affect our results; thus, our analysis is limited by a lack of insight into industrial processes. For example, we assume that chemical conversion rates are at the high end of reported values. Yet if chemical conversion rates were lower – in line with those reported by a recent review of fluorinated refrigerants[26] – then our estimated CFC-113 feedstock production would be higher, bringing it in closer agreement

with reported values (see Supplementary Fig. 1). It is also possible that a temporal increase in conversion rates could account for some portion of the decrease in reported feedstock production, but we do not account for changes in conversion rates in our model. Therefore, the chemical conversion rates are a key source of uncertainty that cannot be resolved without further transparency from the chemical manufacturing industry. In addition, given that we do not have observable proxies for both HFC-125 production pathways, we cannot evaluate which pathway was used or whether temporal or geographic variability in pathway usage contributed to apparent increases in by-product emissions. In particular, the sharp increase in observationally-derived CFC-115 emissions around 2012 could be explained by a shift in production towards the PCE pathway, which produces CFCs as unwanted by-products. This would be consistent with the shift from 4 out of 12 Chinese factories using the PCE pathway in 2011[27] to most global factories using this pathway in 2023[17], but this cannot be confirmed without industry knowledge.

Without additional industry knowledge, it also remains possible that the emissions of CFCs from HFC production are very small, and that the non-bank emissions that we are concerned with come from an unrelated process. In our results, we show that it is possible for HFC production to explain CFC-113, CFC-114, and CFC-115 emissions, but we do not include a term for "unknown or unrelated production" in our simulations, and the previously reported values that inform our priors include combinations of emission and conversion rates that allow HFC production to explain CFC-113, CFC-114, and CFC-115 observations. Previous reports suggest that the large majority of reported CFC-113 produced for use as a feedstock ended up as HFC-134a[15], but this is a qualitative statement that could quantitatively change over time, and given that A 5 countries did not report CFC-113 production, this only pertains to non-A 5 countries. If HFCs are entirely produced by non-CFC production pathways in A 5 countries, then HFC production is not sufficient to explain recent CFC observations and another source of emissions must exist. In particular, recent production of chlorotrifluoroethylene (CTFE) plastics, trifluoroacetic acid (TFA), and the hydrofluoroolefin HFO-1336mzz(Z) were likely to have used CFC-113 or CFC-113a as a feedstock or intermediate[15,38,39]. We assume that emissions from those production processes are negligible here, but future work may have to consider them as the production of those end-products grows.

Uncertainty in global emissions of these compounds and their emission rates also arises from the compounds' lifetimes, which are inversely proportional to emissions in top-down estimates[46]. Observationally derived emissions and simulated mixing ratios were both calculated here using the median of a previously reported most likely lifetime range[42], but different methods for calculating lifetimes yield values that are at least 10% longer or shorter than the lifetimes used here[12,42,46,47]. We test the sensitivity of our model results to CFC lifetimes by simulating mixing ratios using a range of lifetimes informed by previous work, as described further in Supplementary Methods. By doing so, we find that BPE posterior emission rates from feedstock production and consumption vary between 3.1% (2.2–4.2%) and 3.8% (2.9–4.6%) for CFC-113 and between 0.7% (0.3–1.7%) and 1.2% (0.7–2.2%) for CFC-114, and the BPE posterior emission rate for CFC-115 from HFC-125 production varies between 0.7 wt% (0.5–1.0 wt%) and 0.8 wt% (0.5–1.0 wt%). Thus, estimated feedstock and by-product emission rates vary within the 1-$\sigma$ range of estimated uncertainty as the atmospheric lifetimes of these compounds vary within our prescribed range of lifetimes, and adopting these different lifetime values does not qualitatively change the conclusions regarding feedstock reporting.

A final caveat to the assumption that CFC-113 and CFC-114 emissions come from HFC-134a production is that only the minor isomer of CFC-114 (CFC-114a) is required for HFC-134a production. CFC-114a can be produced from CFC-114, or it can be produced directly from CFC-113 or CFC-113a, thereby avoiding the major isomer[26,48]. Thus, it is

possible for CFC-114a to be the only isomer emitted during the production of HFC-134a. Yet, CFC-114a emissions alone cannot explain the increase in the emissions of the sum of the two isomers, so some amount of CFC-114 must be produced and emitted, either as part of the HFC-134a production process or elsewhere. It is also possible that CFC-113a is avoided in the production of HFC-134a (if CFC-113 is converted directly into CFC-114a[48]), but we assume this is unlikely given the enhancement of both CFC-113a and CFC-114a measured in air samples collected downwind of a region where HFC-134a is produced in China[18].

We have developed a Bayesian method that jointly models the production and emission of CFC-113, CFC-114, CFC-115, and HCFC-133a during the chemical manufacturing of HFC-134a and HFC-125. In our model, unintended emissions from these manufacturing processes are able to explain the recent observations of CFC-113, CFC-114, CFC-115, and HCFC-133a that appear inconsistent with the reported production of these compounds. If our assumptions are correct, then this indicates that a growing share of feedstock production is going unreported (possibly due to being considered an intermediate), largely in A 5 countries. We also infer emission rates from facilities around the world that are consistent with best practices, but the added ozone depletion and surface warming potential of these unintended emissions will have to be considered when estimating the total impact of future HFC production nonetheless. This work prompts a broader consideration of the use of regulated substances as feedstocks, including $CCl_4$, and enhances the benefits of compliance with the Kigali Amendment.

## Methods

We extended a previously developed Bayesian model[12,13] to jointly estimate the production for non-feedstock end-uses, production for feedstock use, banks, emissions, and mixing ratios of CFC-113, CFC-114, CFC-115, and HCFC-133a. The modeling approach uses Bayesian Parameter Estimation (BPE), a form of Bayesian analysis that allows us to apply inference to a deterministic simulation model[49,50]. In earlier iterations of the BPE model, the production priors were modeled independently across compounds[12,13]. Here, we have updated the BPE model to explicitly model feedstock production and by-product generation as a function of relevant HFC production in A 5 and non-A 5 countries, thus differentiating emissions by region and accounting for inter-dependencies between the production of these molecules in the manufacturing pipeline.

The BPE model is implemented using the following steps. First, we specify a simulation model of production, banks, emissions, and mixing ratios to jointly represent the manufacturing and emission processes impacting the suite of compounds in our analysis (Eqs. 1–6; input parameters are summarized in Table 3). Next, we develop prior distributions for most of the input parameters to reflect published estimates and their corresponding uncertainties. We then sample from the prior distributions and run the simulation model to obtain a joint distribution of output parameters, including banks, emissions, and mixing ratios. And finally, using Bayes' Rule, we jointly update both input and output parameters given the observed mixing ratios of CFC-113, CFC-114, and CFC-115 from 1990–2020 and observed mixing ratios of HCFC-133a from 1990 to 2019. Global mixing ratios of CFC-113 and CFC-114 have been published through 2020[22] but are not publicly available beyond that year. The period of our analysis is, therefore, 1990–2020. The methods are provided in more detail below.

### Bayesian Parameter Estimation Model
Mixing ratios ($M_{i,t}$) for compound $i$ in time $t$ are simulated as

$$M_{i,t+1} = M_{i,t} * e^{-\tau_i^{-1}} + A * E_{i,t}, \tag{1}$$

where $A$ is a constant that converts the mass of emissions (Gg) into mixing ratios (ppt) and accounts for the discrepancy between surface and global mean atmospheric mixing ratios[10]. As mixing ratios in year $t+1$ depend on processes in year $t$, this formulation assumes a 1-y mixing time in the troposphere[51]. Previously reported atmospheric lifetimes ($\tau_i$) of 93, 191, 540, and 4.6 years were used for CFC-113, CFC-114, CFC-115, and HCFC-133a, respectively[42,52]. The lifetimes for CFC-113 and CFC-114 used here are for the dominant isomer of these compounds and therefore overestimate the total lifetime of the sum of the isomers (lifetimes of CFC-113a and CFC-114a are 55 and 105 years, respectively[53]). If the atmospheric abundance of minor isomers was significant, then posterior estimates would be biased towards simulations with lower total emissions. However, atmospheric mixing ratios of CFC-113a and CFC-114a were 1.0 ppt and 1.1 ppt in 2020[22], while atmospheric mixing ratios of the sum of CFC-113 and CFC-114 isomers were 69.4 ppt and 16.3 ppt, respectively[54], so we assume this bias is small.

To simulate the emissions time series used in Eq. 1, we summed the four emission sources that we assume comprise the total emissions of each compound. These are: production for non-feedstock use ($Prod_{i,j,t}$, where $j$ denotes use in a short-lived or long-lived bank), banks ($B_{i,j,t}$), use as a feedstock in HFC-134a production ($FS_{i,t}^k$, where $k$ denotes use in A 5 or non-A 5 countries), and generation as a by-product during the manufacturing of HFC-125 ($HFC125_t^k$). Note that $Prod_{133a,j,t} = 0$ and $B_{133a,j,t} = 0$ for all $t$, as non-feedstock production of HCFC-133a was not reported. In addition, $FS_{115,t}^k = 0$ for all $t$, as CFC-115 is not used as a feedstock in manufacturing HFC-134a. Following from previous work[13], we assume two categories for non-feedstock end-uses with distinct emission rates: short-lived banks and long-lived banks. For each category, the fraction of production emitted directly (i.e., the direct emission rate) is denoted by $DE_{i,j}$ and the fraction of the bank released each year (i.e., the release fraction) is denoted by $RF_{i,j}$. Feedstock and by-product emission rates for each country classification are denoted by $FE_i^k$ and $BP_i^k$. $DE_{i,j}$, $RF_{i,j}$, $FE_i^k$, and $BP_i^k$ are all assumed to be constant with time. Thus, the emission time series for each compound is calculated as

$$\begin{aligned} E_{i,t} = & \sum_{j=1}^{N_1}(DE_{i,j} * Prod_{i,j,t} + RF_{i,j} * B_{i,j,t}) \\ & + \sum_{k=1}^{N_2}(FE_i^k * FS_{i,t}^k + BP_i^k * HFC125_t^k), \end{aligned} \tag{2}$$

where direct and bank emissions are summed over $N_1$ bank categories (i.e., long and short banks), and feedstock and by-product emissions are summed over $N_2$ country classifications (i.e., A 5 and non-A 5).

Banks are simulated recursively for each bank category, $j$, as

$$B_{i,j,t+1} = (1 - RF_{i,j}) * B_{i,j,t} + (1 - DE_{i,j}) * Prod_{i,j,t}, \tag{3}$$

and feedstock production in each country classification is calculated as

$$\begin{aligned} FS_{114,t}^k = & M_{114}/M_{134a} * \chi_t^k * HFC134a_t^k \\ & * 1/(C_{114 \to 134a} * (1 - FE_{114}^k)), \end{aligned} \tag{4}$$

$$\begin{aligned} FS_{113,t}^k = & M_{113}/M_{114} * FS_{114,t}^k \\ & * 1/(C_{113 \to 114} * (1 - FE_{113}^k)), \end{aligned} \tag{5}$$

$$\begin{aligned} FS_{133a,t}^k = & M_{133a}/M_{134a} * (1 - \chi_t^k) * HFC134a_t^k \\ & * 1/(C_{133a \to 134a} * (1 - FE_{133a}^k)), \end{aligned} \tag{6}$$

where $M_i$ is the molar mass of compound $i$, $\chi_t^k$ is the fraction of HFC-134a produced via the PCE pathway (which may emit CFC-113 and CFC-114), and $C_{a \to b}$ is the conversion rate from compound $a$ to compound $b$. $\chi_t^k$ thus represents the dependencies between CFC-114 and HCFC-133a

**Table 3 | Parameters used in the simulation model described by Eqs. 1–6**

| Parameter | Description | Units |
|---|---|---|
| $A$ | Conversion factor between the mass of emissions and global mixing ratios | $ppt \cdot Gg^{-1}$ |
| $\tau_i$ | Atmospheric lifetime of compound $i$ | y |
| $Prod_{i,j,t}{}^*$ | Production of compound $i$ for non-feedstock end-use $j$ in year $t$ | kg |
| $DE_{i,j}{}^*$ | The emission rate of compound $i$ from non-feedstock end-use $j$ during the year of production | $kg \cdot kg^{-1}$ |
| $RF_{i,j}{}^*$ | Release rate of compound $i$ from bank type $j$ | $y^{-1}$ |
| $FS_{i,t}^k$ | Production of compound $i$ for use as a feedstock in country classification $k$ in year $t$ | kg |
| $FE_i^{k*}$ | Emission rate of compound $i$ from use as a feedstock in country classification $k$ | $kg \cdot kg^{-1}$ |
| $HFC125_t^k$ | Production of HFC-125 in country classification $k$ in year $t$ | kg |
| $BP_i^{k*}$ | The by-product emission rate of compound $i$ from the production of HFC-125 in country classification $k$ | $kg \cdot (kg\ HFC\text{-}125)^{-1}$ |
| $M_i$ | The molar mass of compound $i$ | $g \cdot mol^{-1}$ |
| $HFC134a_t^k$ | Production of HFC-134a in country classification $k$ in year $t$ | kg |
| $\chi_t^{k*}$ | Fraction of HFC-134a produced via the PCE pathway in country classification $k$ in year $t$ | $kg \cdot kg^{-1}$ |
| $C_{a \rightarrow b}$ | Chemical conversion rate from compound $a$ to compound $b$ | $kg \cdot kg^{-1}$ |

Parameters that are updated by the BPE model are marked with an asterisk; prior and posterior distributions for these parameters are shown in Supplementary Figs. 3–5.

feedstock production, and $C_{113 \rightarrow 114}$ represents dependencies between CFC-113 and CFC-114 feedstock production in the deterministic simulation model. For simplicity, we assume that all feedstock and by-product emissions occur in the same year that the feedstocks and by-products are produced.

### Prior distributions

Non-feedstock production priors, $Prod_{113,j,t}$, $Prod_{114,j,t}$, and $Prod_{115,j,t}$ were developed for years prior to 1989 using production data reported to Alternative Fluorocarbons Environmental Acceptability Study (AFEAS)[55]. For CFC-113, this data was augmented according to the WMO (2003) correction[47], and total production data from 1989–2016 were taken from the WMO 2022 report on production and consumption of ozone depleting substances[23]. We assume no production following the end of reporting. For CFC-114 and CFC-115, total production data from 1989–2003 were taken as the greater of AFEAS data or AFEAS data scaled to match WMO production data, and total production data from 2004–2019 were taken from WMO's 2022 report[23]. To account for uncertainty in reported production, we assume lognormal distributions for $Prod_{113,j,t}$, $Prod_{114,j,t}$, and $Prod_{115,j,t}$, following previous work[11], where we assume the bias in reported data has a correlation term, $\rho_{i,j}$, that we infer in the BPE model (see ref. 11 for more details). We set lower bounds of these distribution as 70%, 95%, and 80% of reported values, respectively, to ensure that observed mixing ratios were within the simulated priors[12]; see ref. 11 for further description of these distributions.

The allocation of production to short or long bank equipment types for CFC-113 and CFC-114 was informed by AFEAS data when available and fixed to values from the final year of AFEAS data afterwards. Given the poor fit between simulated mixing ratios and observations that was previously reported for CFC-115[13], we set the fraction of CFC-115 production allocated to short banks as an uncertain parameter with a prior uniform distribution between 50–90%. This uncertain parameter reflects uncertainty in AFEAS production allocation for CFC-115 – only production for refrigeration (i.e., long bank) was reported to AFEAS, but CFC-115 was also used as an aerosol propellant (i.e., short bank)[56], though this was not documented in AFEAS data. The addition of this parameter resulted in an improved fit between posterior simulated mixing ratios and observations (Supplementary Fig. 2), so we continued with this altered end-use allocation.

Production of HFC-134 and HFC-125, $HFC134a_t^k$ and $HFC125_t^k$, from 1990 to 2019 were taken from a previously reported joint bottom-up and top-down estimate[25] and were assumed to be 0 prior to 1990. These data are calculated using data from several sources, including consumption reported by non-A 5 countries to the United Nations Framework Convention on Climate Change[57], previously estimated Chinese and Indian consumption estimates[58,59], and emissions inferred from AGAGE[60] and National Oceanographic and Atmospheric Administration (NOAA) Global Monitoring Laboratory[61] observations of surface mixing ratios. Values are reported for A 5 and non-A 5 countries, thereby allowing for separation of production from the two classifications in our simulations. Note that we do not account for uncertainty in HFC production in our model, as uncertainties in the $FE_i^k$ and $BP_i^k$ terms in Eq. 2 and $\chi_i^k$ term in Eqs. 4 and 6 and would linearly compensate for biases in HFC production. Nonetheless, we note the posterior distributions of these terms are conditional on the adopted HFC production time series.

Prior distributions of $DE_{i,j}$ and $RF_{i,j}$ were informed for each non-feedstock end-use by industry-reported data[62], following recent work[12]. $FE_i^k$ distributions were informed by the range of likely values reported by MCTOC[16] (1.5–6.1%). For computational efficiency, after simulating each gas independently, the $FE_{114}^k$ and $FE_{133a}^k$ parameter spaces were updated to remove the tails of the parameter space where the conditional probability of the data given the parameter value was near zero. As $FE_{114}^k$ and $FE_{133a}^k$ posteriors suggested values lower than the MCTOC range, these distributions were also adjusted to include values between 0 and 1.5%. $BP_i^k$ distributions were informed by a recent patent for HFC-125 production that reports by-product generation rates relative to HFC-125 production[30], with maximum emission rates of 2% and 1.5% for CFC-115 and HCFC-133a, respectively, and CFC-113 and CFC-114 emission rates of no more than half of the CFC-115 emission rate. We do not know how much of each by-product is emitted (as opposed to captured and/or destroyed), so we assumed beta distributions with parameters (2, 2) for $FE_i^k$, $BP_{115}^k$, and $BP_{133a}^k$ priors and uniform distributions between $0 – 0.5 * BP_{115}^k$ for $BP_{113}^k$ and $BP_{114}^k$ priors. Previous work has assumed that emission rates from chemical manufacturing are higher in A 5 countries than in non-A 5 countries[41]; to explore this possibility, we specify independent but identical priors for $FE_i^k$ and $BP_i^k$ for A 5 and non-A 5 countries.

Following a series of patents in which the chemical conversion rates of CFC-113 to CFC-114, CFC-114 to HFC-134a, and HCFC-133a to HFC-134a are reported under various conditions[26,28,36,37], $C_{113 \rightarrow 114}$, $C_{114 \rightarrow 134a}$, and $C_{133a \rightarrow 134a}$ were set to fixed values of 98% and 94%, and 95%, respectively. Although we do not know which catalysts and

reaction conditions are used, we assume that conversion rates are at the high end of reported values based on the assumption that this is a mature industry where manufacturers would want to minimize unused resources. We set these as fixed values as the technology for these chemical conversion processes is not known to have changed with time. Simulations run with lower conversion rates suggest greater feedstock production, but this does not qualitatively change our conclusions (i.e., inferred under-reporting of feedstock production in A 5 countries is increased when conversion rates are lowered, so our choice of conversion rates makes our unreported feedstock production results conservative). Prior distributions for $FE_i^k$, $BP_i^k$, and $C_{a \to b}$ are summarized in Supplementary Table S1.

$\chi_t^k$ was assumed to be a uniform distribution between 0–70%, based on previous reporting that the TCE pathway is more commonly used for HFC-134a production[31–33]. This prior incorporates an autocorrelation term that is sampled from a uniform distribution between 0.95–1.0 to reflect the potential for gradual change to global manufacturing.

As the initial year of reporting varies, we start our simulation model in 1955 for CFC-113, 1935 for CFC-114 and CFC-115, and 1990 for HCFC-133a. Initial mixing ratios are assumed to be 0 for CFCs and 0.0489 ppt for HCFC-133a[34]. As available production data for our bottom-up emissions estimates end in 2019, we implement the simulation model out to 2020.

## Likelihood function

As in previous work[12], the difference between modeled and observed mixing ratios was assumed to be normally distributed with a mean of zero. Therefore, the likelihood function is a multivariate normal likelihood function of the difference between modeled and observed mixing ratios:

$$P(D_i|\theta) = \frac{1}{\sqrt{(2\pi)^{N_{obs}}|\Sigma_i|}} e^{\left(-\frac{1}{2}\epsilon_i^T \Sigma_i^{-1} \epsilon_i\right)}, \tag{7}$$

where $D_i$ is a vector of annual global mean observed mixing ratios for each year from 1990–2020, $N_{obs}$ is the length of $D_i$ ($N_{obs}$ = 31 for CFCs and 6 for HCFC-133a, see below), $\theta$ is the vector of all input and output parameters from the simulation model, $\epsilon_i$ is an $N_{obs}$x1 vector of the difference between modeled and observed mixing ratios in each year with a temporal covariance matrix $\Sigma_i$. As mixing ratios are constrained through 2020, emissions can only be constrained through 2019 (see Eq. 1).

Within the error covariance matrix, we assumed additive error in uncertainties for each compound. Therefore, $\Sigma_i$ contains the sum of the uncertainties in observed and simulated mixing ratios along its diagonals with the off-diagonals autocorrelated with coefficient of 0.95, representing an expected high autocorrelation in error for both the observed and simulated mixing ratios. Based on uncertainties in measurements and the relationship between surface point observations and global mean mixing ratios, CFC-113, CFC-114, and CFC-115 global mixing ratios have uncertainties of 1.5, 3.0, and 3.0%, respectively[46]. The uncertainty in the simulation model is not known, and due to computational limitations, sampling model uncertainties in the joint BPE model was not feasible. We therefore iteratively selected model uncertainties for each compound by initially specifying a prior model uncertainty error as a function of observed mixing ratios. We then ran the BPE model for each compound independently and selected the most likely model uncertainty term, with a precision of 0.5% of observed mixing ratios. This resulted in total uncertainties of 2.0% of observed mixing ratios for CFC-113 and 4.0% for CFC-114 and CFC-115. For HCFC-133a, measurements had an estimated 2-$\sigma$ uncertainty of 10%[34], and given that our assumptions do not capture variability in industrial practices that have previously been hypothesized to result in variability in HCFC-133a emissions[21], we aggregated the

observational data into 5-y annual means and adopted a total uncertainty of 20% of observed mixing ratios for HCFC-133a. This measurement uncertainty, level of temporal aggregation, and potential interannual variability that is not captured by our model also cause the off-diagonal autocorrelation term within $\Sigma_{133a}$ to be uncertain and lower than that of the CFCs. Therefore, we modeled this autocorrelation term as a beta distribution between 0.6–0.8 with parameters (2, 2). We tested the sensitivity of our results to the model uncertainties by evaluating the likelihood function with uncertainties 50% smaller and 25% larger than those listed here, and the results were not qualitatively impacted.

Global mean mixing ratios were estimated by the AGAGE 12-box model of atmospheric transport[46,63] using measurements taken by the AGAGE surface observation network[27,60]. HCFC-133a data were taken from a previously published work[21] that followed this method.

We tested the robustness of our results to a different observational dataset for CFC-113 from the NOAA network[61] in place of AGAGE observations. CFC-114, CFC-115, and HCFC-133a are not measured by the NOAA network and, therefore, were unchanged in this sensitivity test. Posterior estimates of feedstock and by-product emission rates calculated with AGAGE and NOAA datasets are within 1-$\sigma$ uncertainty (Supplementary Table S2), indicating that our results are not specific to our choice of observational data.

## Estimation of posterior distributions

To estimate the joint posterior distributions of the input and output parameters of Eqs. 1–6, we implement Bayes' Rule:

$$P(\theta|D_{113}, D_{114}, D_{115}, D_{133a}) \propto$$
$$P(\theta)P(D_{113}|\theta)P(D_{114}|\theta)P(D_{115}|\theta)P(D_{133a}|\theta), \tag{8}$$

where $\theta$ denotes the input and output parameters of the deterministic simulation model (Eqs. 1–6), and thus $P(\theta)$ denotes the joint prior distribution of the input and output parameters. $D_i$ denotes the observed mixing ratios of molecule $i$. As in previous work[12], we assume that the data ($D_{113}, D_{114}, D_{115}, D_{133a}$) are conditionally independent given $\theta$, and that $P(D_i|\theta)$ is the multivariate likelihood function of all years of observed mixing ratios for molecule $i$ given $\theta$. In addition, for computational efficiency, Eq. 8 is estimated through sequential Bayesian updating in three steps. We first update the input parameters given $D_{115}$:

$$P(\theta|D_{115}) \propto P(\theta)P(D_{115}|\theta). \tag{9}$$

The posterior $P(\theta|D_{115})$ distribution is then used as the prior and updated given $D_{114}$ and $D_{133a}$:

$$P(\theta|D_{114}, D_{115}, D_{133a}) \propto P(\theta|D_{115})P(D_{114}|\theta)P(D_{133a}|\theta). \tag{10}$$

This posterior is then updated once more given $D_{113}$ to obtain the full joint posterior:

$$P(\theta|D_{113}, D_{114}, D_{115}, D_{133a}) \propto$$
$$P(\theta|D_{114}, D_{115}, D_{133a})P(D_{113}|\theta). \tag{11}$$

For further description on the implementation of the BPE model, see ref. 12.

The posterior distribution was estimated using the sampling importance ratio (SIR) method[50,64,65], which involves first sampling the prior distributions and then resampling the prior samples at a rate proportional to the importance ratio, which is proportional to the likelihood function defined in the previous subsection. As noted previously, we implement SIR through sequential updating. To do so, we first solve Eq. 9 by sampling 2,000,000 samples from $\theta$'s prior distribution and run the simulation model for CFC-115. Note that for

computational efficiency in the first iteration of sequential updating, we only sample the parameters that are used in the CFC-115 simulation model. We then resample 1,000,000 samples from these prior samples, proportional to each sample's importance ratios, given by

$$\frac{P(\theta|D_{115})}{P(\theta)} \propto P(D_{115}|\theta). \quad (12)$$

Of all the parameters in the CFC-115 simulation model conditionally dependent on $D_{115}$, $BP_{115}^k$ is the only one that informs priors for CFC-113 and CFC-114, and thus HCFC-133a as well. In the second iteration of sequential updating, the posterior samples of $BP_{115}^k$ are used to inform the priors of $BP_{113}^k$ and $BP_{114}^k$. For all other parameters in $\theta$ used in the CFC-114 and HCFC-133a simulation models, we sampled from their priors 1,000,000 times and ran the simulation model for CFC-114 and HCFC-133a. All 1,000,000 samples (i.e., both the updated parameters from the CFC-115 simulation and the prior samples from CFC-114 and HCFC-133a) were then resampled 300,000 times, proportional to the importance ratio:

$$\frac{P(\theta|D_{115}, D_{114}, D_{133a})}{P(\theta|D_{115})} \propto P(D_{114}|\theta)P(D_{133a}|\theta). \quad (13)$$

In the final sequence of updating, the $FS_{114}^k$ posterior is used to inform the $FS_{113}^k$ prior (Eq. 5). We drew 300,000 samples from all remaining parameters in $\theta$ and ran the CFC-113 simulation model. Finally, to obtain the full joint posterior distribution, all 300,000 samples (i.e., the updated parameters from the CFC-114, CFC-115, and HCFC-133a simulations and the prior samples from the CFC-113 simulation) are resampled 100,000 times proportional to the importance ratio:

$$\frac{P(\theta|D_{115}, D_{114}, D_{133a}, D_{113})}{P(\theta|D_{115}, D_{114}, D_{133a})} \propto P(D_{113}|\theta). \quad (14)$$

### Ozone depletion and global warming potentials

To quantify how HFC-134a and HFC-125 production may delay the healing of the ozone layer and warm Earth's surface, we weight emissions attributed to the production of these compounds by the ozone-depleting potential (ODP) and 100-year global warming potential (GWP) of compounds being emitted. HFCs do not contribute to ozone destruction, so the ODP of unintended feedstock and by-product emissions constitutes the entire ODP attributable to HFC-134a and HFC-125. For GWP, we included the contribution of HFC-134a and HFC-125 emissions[25] (GWPs of 1300 and 3170, respectively). These observationally-derived emissions estimates can only account for what has been emitted (either directly from the production process or from banks) and cannot capture the GWP of HFCs currently banked that may leak from their current reservoir until the end of their equipment's life.

For CFC-113 and CFC-114, which each had two isomers emitted from 2004–2019, we weighted ODPs and GWPs based on the recently reported isomeric composition of emissions. Emissions of the minor isomers, CFC-113a and CFC-114a, averaged 2.0 Gg · y$^{-1}$ and 0.45 Gg · y$^{-1}$ from 2004–2019[22], respectively, making them both roughly 40% of the total CFC-113 and CFC-114 emissions. Using previously reported ODPs of 0.82, 0.73, 0.53, and 0.72 and GWPs of 6530, 3930, 9450, and 7410 for CFC-113, CFC-113a, CFC-114, and CFC-114a[66], respectively, we calculated weighted ODPs and GWPs of 0.78 and 0.61 and 5490 and 8634 for CFC-113 and CFC-114, respectively. For CFC-115 and HCFC-133a, we used previously reported ODPs of 0.45 and 0.019 and GWPs of 9630 and 378[66]. We report values in units of ODP-Gg, which is the mass-weighted equivalent emissions of CFC-11, and TgCO$_2$eq, which is the mass of CO$_2$ that would result in the same radiative forcing on a 100-year time scale.

## Data availability

The mixing ratio, production, and emissions data generated and/or analyzed in this study have been deposited on Zenodo[67].

## Code availability

The code for Bayesian analysis and the plots presented in this work are available on Zenodo[67]. All analysis was done in MATLAB.

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

## Acknowledgements

The authors would like to acknowledge support from the VoLo Foundation and from the Atmospheric Chemistry Division of the National Science Foundation (grant no. 2128617, M.L.). The authors would also like to thank Luke Western for providing output from the AGAGE 12-box model, as well as Susan Solomon and Stefan Reimann for helpful discussions.

## Author contributions

S.B. and M.L. conceptualized the work and developed the methods. S.B. conducted the analysis, interpreted the data, and drafted the manuscript. S.B. and M.L. contributed revisions to the manuscript.

## Competing interests

The authors declare no competing interests.
