## [Transparent Peer Review file · Nature Communications]

Bayesian modeling of HFC production pipeline suggests growth in unreported CFC by-product and feedstock production

Corresponding Author: Dr Stephen Bourguet

Version 0:

Reviewer comments:

Reviewer #1

(Remarks to the Author)

This paper uses a Bayesian inference model to simulate the production of HFCs and quantify the associated emissions of CFCs in those processes. Since the discovery of atmospheric increases in a few very minor CFCs the cause has been a puzzle. By-product and feedstock emissions in the manufacture of other gases have been suggested but the community has been lacking quantitative modelling to confirm the importance. This paper provides that.

I think that this is an interesting and useful paper for the community. It advances the topic by providing a framework to quantify the fugitive CFC emissions. The paper provides quantification of the emissions from HFC production which also then points to additional negative consequences of their manufacture.

The paper is very detailed with a lot of quantitative discussion. That is understandable given the nature of the work and topic. However, because of that some top level messages about the importance of the work, e.g. improved understanding of the sources of the CFCs or elimination of other hypotheses, is lost.

As an atmospheric science paper I only have some minor comments. I must admit I am not expert in details of Bayesian modelling. I found that the paper explained the method, input data, assumptions and caveats fairly well so that others working in this field can test the quantitative results.

Specific comments:

Line 22. You need to explain 'Article 5' if used in the abstract.

Line 22 'atmospheric growth rates of CFC-113 and CFC-114'. CFC-113 is not increasing in the atmosphere so this sentence is wrong and confusing. (I realise that the isomer CFC-113a is increasing but some editing is needed).

Lines 25-26 'underscores the importance....'. I can see the point being made but I think the text is vague. If you mean that the by-product emission of CFCs is another reason for controlling HFCs then please say so explicitly.

Line 29. 'radiative forcings'. This is the wrong term. My understanding is that Radiative Forcing is used to mean the actual change in the climate forcing due to a gas since the pre-industrial period. That depends on the radiative properties and the change in abundance. You mean something like GWP here.

Line 30. ODP = ozone depletION potential

Line 44. UNEP = United Nations Environment Programme, i.e. not 'environmental' and with normal (non-American) spelling.

Line 77. Typo 'hydrofluorocarbons'.

Figures: All of the figures need a lot of work in my opinion. The panels are small and difficult to read. For the draft they could have used more of the width of the paper. There is also quite a lot of white space in between some panels. There is a lack of tick marks (and even axes) on some plots (e.g. Figure 3).

Line 116. This note about the two isomers of CFC-113 should be made and explained earlier.

Figure 6 and estimate of equivalent ODP emissions. Panel B shows that the major contributor is CFC-115 due to the larger emissions in HFC-125 production and the ODP value of 0.45. However, CFC-115 has a lifetime of 540 years, which is much longer than CFC-113 etc, and also very long compared to the timescale for expected recovery of the ozone layer. The CFC-115 emitted now will in effect give a very slow (small) but long-term depletion of ozone (although a question is how ODP values will differ over such a long timescale). I realise that this is complex and raises a number of issues but I don't think it makes sense to calculate a simple ODP-weighted metric without some comment and maybe also present the results without CFC-115.

(Remarks on code availability)

Reviewer #2

(Remarks to the Author)

Comments on "Bayesian modeling of HFC production pipeline suggests growth in unreported CFC by-product and feedstock production" by Bourguet and Lickley

Evaluation of trends in emissions of controlled substances by the Montreal Protocol with a view to identifying currently unknown emission sources is important, topic and of broad interest. The authors developed a Bayesian framework to quantify emissions of CFCs in HFC production, to explain discrepancies between emissions derived from reported information and observed atmospheric CFCs concentrations. They found the use of CFCs as feedstocks in the HFC production accounts for most of the CFCs emissions under report. Thus, they speculate that there is un-reported feedstock production in A5 countries, which may explain recent unexpected atmospheric growth in CFCs concentrations.

Major comments:

The results of this paper stand on the assumptions that the HFC production is accurate and that non-A5 countries produced and consumed CFC-113 and CFC-114 feedstocks in separate processes. However, I have large concern on both assumptions. The authors also acknowledged some of the uncertainties after the conclusions in section 3.1. Therefore, I suggest the authors to be cautious on the speculations that are made in section 3.1. In terms of the second assumption, the authors showed the consistency between the reported values by non-A5 countries and their modeling results as a validation of the assumption, but this is not robust.

Figure 4B compares the reported production of CFC-113 (by non-A5 countries) with the BPE estimations for non-A5 countries. The two have a general consistency as most of the reported values falls within the 1-sigma interval of the BPE estimations, but the recent reported increase from 2015-2019 of CFC-113 production by non-A5 countries are not capture by the BPE estimations. This is a flaw that undergrade the confidence of the conclusion (the authors made) that recent increase in CFC-113 is largely from A5 countries.

There is a time lag between the reported values and emissions/atmospheric concentrations considering implementations and atmospheric transport. What is the time the authors assumed for the reporting and atmospheric mixing? I suggest the authors to extend their results to the most recent years, e.g. 2023. What about the most recent reported values by non-A5 countries? It could simply be the continuing relatively-high-level production (~ 100 Gg/year) by the non-A5 countries.

Another issue is about the inconsistency between the BPE posteriors with observationally derived emissions of CFCs (Figure 3 right column). I think there is no significant increase in the observational derived emissions, especially for CFC-113 and HCFC-133a, as the interannual variability is large. The BPE posterior clearly did not capture the large interannual variability. The authors claim they "do not impact variability beyond interannual timescales". Is it true? The unexplained interannual variability adds to the uncertainty in long-term change, leading to a low confidence in the increase trend showed by BPE posterior emissions. It is especially concerning when the studied period is relatively short (16 years from 2004 to 2019).

Specific comments:

Lines 12-14: Can you make this clearer? For example, adding something like "increasing" or their positive trend or increase rates here. What are the values of "expected given global reporting"?

Line 28, Change "When" to "Once"

Figure 1: Can you add some explanation about the possible causes of annual variation of CFC-113 in Figure 1A, while less variability for CFC-114 and CFC-115?

Lines 169-177, Can you explain how these values are calculated? They are hardly to be estimated from Figure 3.

Lines 199-202: Note the uncertainty is very large. According to the BPE priors, the CFC-113 and CFC-114 production in A5 countries changes from 9-47% (2004-2008) to 35-80% (2015-2019). Is the "increase" statistically significant?

Tables 1 and 2, Can you also include the emission range here?

I am not sure it is a good idea to estimate the global warming potentials for the future scenarios because there are many assumptions used.

Figures 1, 3, 5, 6: The end year "2019" should be noted on the time axis.

(Remarks on code availability)

I have downloaded them but have not run the code since it is written in MATLAB.

Reviewer #4

(Remarks to the Author)

This manuscript improves our understanding of the increasing emissions of several CFCs, by examining their emission breakdowns from allowed feedstock use, by-product emissions and consumption/banks etc., following a previously developed Bayesian inference framework. The study throws light on the exempted usage of ODSs under the Montreal Protocol, and by-product emissions of ODSs, which are somehow ignored and could be potential remaining challenges of the ozone layer recovery, and provides useful tool for potential further studies. The authors also acknowledge the limitations and uncertainties within the modeling framework of the study. The manuscript is generally well-structured and well-written. I believe the manuscript is suitable for publication in Nature Communications, after some revisions.

My specific comments are below:

1. I think the authors need to present a comparison of the posterior values with prior values of each variable included in the Bayesian, for a complete picture of the performance of Bayesian. It is unclear now whether some of the conclusions are due to the constraint by the observations, or due to the prior values.
E.g. Lines 203ff. The posterior emission rate seems to be close to the mean of the prior distribution. It is not clear whether the posterior emission rates are due to constraint, or prior. I would suggest adding a comparison of the fit to observations using prior and posterior parameters.
Specify what variables were constrained in Methods/paragraph 112ff.
Tabulate the input parameters and the definitions of all non-feedstock end uses.
Line 527ff: Why the prior FE is 0-4% lower than the 1.5-6.2%? what is the distribution for prior feedstock emission rate?
Section 2.2-2.3. I suggest the authors also discuss the annual changes in the posterior feedstock/byproduct emission rates.
2. Some of the wording needs some clarification.
E.g. both "feedstock usage", "feedstock production" and "feedstock" appear in the text. I understand the feedstock production is used in the Bayesian calculation for this process, but I think the authors need to clarify the statement of feedstock production/consumption/usage and where emissions will happen related to this process somewhere in the text.
E.g. Line 174 "HFC-125 production" -> "byproduct emissions from HFC-125 production". Same for 176, 177 and elsewhere in the text. Always specify the emission information in the text to make things clear.
E.g. Line 241: "CFC-113 and CFC-114 emission rates" -> "byproduct emission rates during HFC-125 production"
3. The last sentence of abstract: "underscores the importance of the HFC production phasedowns...". The major contribution from phasing down HFC production is HFC itself (from HFC consumption), thus not a finding of this study. I suggest reconsidering this statement.
4. Reference 1 - Suggest referring to the specific chapter of the Ozone Assessment report to credit the chapter authors, as done in ref 23, 7, 64.
5. Figure 1 caption: The authors use "portion" here and elsewhere several times. I think using portion (usually for percentage) to represent the absolute emissions may lead to some ambiguity.
6. Line 124: "in their production", what does "their" refer to?
7. Line 126: "feedstock production and by-product emission rates", of the CFCs? And I think the feedstock production are both constrained?
8. Fig 3. You need to explain the legend in the figure caption, such as "no HFC production". "Production" should be non-feedstock production.
9. The uncertainties ranges in the text and tables: the authors should explain what the uncertainties are, especially for the average values over a time period. The description in Table 1 looks a bit confusing.
10. Line 228: why using "observationally-derived" emissions here not the BPE estimated emissions?
11. Section 2.4 and corresponding Methods part: I believe that ODP and GWP terms are a fix term for a substance representing their capacity of depleting the ozone layer and radiative forcing. I do not think the use of "ODP of emissions" is appropriate. It is confusing especially you use "ODP" for both actually ODP and ODP-eq emissions in the text. The ODP of emissions -> The ODP-weighted emissions or ODP-Gg emissions.
Also, define "unintended" here.

12. Lines 350ff: however, your conclusion from the posterior emission rate in the above section (lines 203ff) suggest that: the posterior feedstock emission rate is low and not supporting the transport emissions between production and consumption. Does it mean that your emission rate results support that CFCs are used as intermediate and not subject to reporting both in A5 and non-A5 countries?

13. Eq(6), should be FE133a. Missing "a".

14. Line 554-556. What is the "autocorrelation term" here?

15. Line 585-586. What is the 0.6-0.8 here and how it compares to the 0.95 in Line 571? Have the authors considered a year-decay correlation coefficient?

(Remarks on code availability)

The information of the Code is for "PNAS submission". Not sure if this should be corrected.

Version 1:

Reviewer comments:

Reviewer #2

(Remarks to the Author)

Comments on "Bayesian modeling of HFC production pipeline suggests growth in inreported CFC by-product and feedstock production" by Bourguet and Lickley

Thank the authors for having addressed some initial concerns and comments in the revised version. Some important clarifications have been added. However, there are still a few major points outlined below which required to be addressed to ensure the article is ready for publication.

1. Figure 1 appears improved by smoothing, but it primarily reveals a high level of uncertainty relative to a minor increase, particularly for CFC-113. This suggests that the observed increase is likely not significant. The authors claim, 'CFC-114, CFC-114, and CFC-115 emissions ... increased between 2004 and 2019.' However, is this increase truly significant? Given the high uncertainty and limited sample size, any confident conclusions regarding long-term changes are questionable and may reflect a fundamental flaw in the analysis. In the Line 84, it mentions that "to 500 Gg/year in 2019 shown in Fig.1C-D", this number is the sum from Fig1C and Fig1D?

2. Similarly, for Figure 3 and any references to "trend" elsewhere in the text, The authors still need to conduct a statistical test to substantiate these trends.

3. Line 223 (in track-change version): "Notably, reported CFC-113 production values are greater than our 1-sigma interval for estimated non-A5 production from 2004-2012 and at the high end of the interval from 2013-2019. Given that reported values for CFC-113 came only from non-A5 countries from 2008-2019, this suggests that feedstock production in these countries was likely not underreported during this time". The statement is reasonable. However, it could also imply that the estimated CFC-113 production values might underestimate actual production in non-A5 countries, thereby mistakenly overestimating the contribution from A5 countries.

4. The authors need to double-check the Global production of CFC-113 and CFC-114 in Fig4B and Figure 4C, it seems by eyes that Fig4B of CFC-113 was wrongly plotted which is not consistent with the line Lin 196 (to 205 Gg/y in 2019".

(Remarks on code availability)

They can be downloaded but have not tested since I am not a matlab user

Reviewer #3

(Remarks to the Author)

(Remarks on code availability)

Reviewer #4

(Remarks to the Author)

The authors have addressed my concerns well.

(Remarks on code availability)

Reviewer #1 (Remarks to the Author):

This paper uses a Bayesian inference model to simulate the production of HFCs and quantify the associated emissions of CFCs in those processes. Since the discovery of atmospheric increases in a few very minor CFCs the cause has been a puzzle. By-product and feedstock emissions in the manufacture of other gases have been suggested but the community has been lacking quantitative modelling to confirm the importance. This paper provides that.

I think that this is an interesting and useful paper for the community. It advances the topic by providing a framework to quantify the fugitive CFC emissions. The paper provides quantification of the emissions from HFC production which also then points to additional negative consequences of their manufacture.

The paper is very detailed with a lot of quantitative discussion. That is understandable given the nature of the work and topic. However, because of that some top level messages about the importance of the work, e.g. improved understanding of the sources of the CFCs or elimination of other hypotheses, is lost.

As an atmospheric science paper I only have some minor comments. I must admit I am not expert in details of Bayesian modelling. I found that the paper explained the method, input data, assumptions and caveats fairly well so that others working in this field can test the quantitative results.

We thank Reviewer 1 for their comments and constructive feedback.

Specific comments:

Line 22. You need to explain ‘Article 5’ if used in the abstract.
A parenthetical with “low- to middle-income” has been added.

Line 22 ‘atmospheric growth rates of CFC-113 and CFC-114’. CFC-113 is not increasing in the atmosphere so this sentence is wrong and confusing. (I realise that the isomer CFC-113a is increasing but some editing is needed).

This sentence has been changed to (Lines 21–23):

“Our results suggest that unreported feedstock production in Article 5 (low- to middle-income) countries may explain the unexpected persistence of the atmospheric mixing ratios of CFC-113 and CFC-114”

Lines 25-26 'underscores the importance...'. I can see the point being made but I think the text is vague. If you mean that the by-product emission of CFCs is another reason for controlling HFCs then please say so explicitly.

We've changed this to be more specific (Lines 26 - 28):

“Nonetheless, this work demonstrates the environmental impacts of tightened ODS feedstock regulations and adds a reduction in CFC emissions to the benefits of the HFC production phasedowns scheduled by the Kigali Amendment.”

Line 29. 'radiative forcings'. This is the wrong term. My understanding is that Radiative Forcing is used to mean the actual change in the climate forcing due to a gas since the pre-industrial period. That depends on the radiative properties and the change in abundance. You mean something like GWP here.

This has been changed to “global warming potentials”

Line 30. ODP = ozone depletION potential

This has been fixed.

Line 44. UNEP = United Nations Environment Programme, i.e. not 'environmental' and with normal (non-American) spelling.

This has been fixed.

Line 77. Typo 'hydrofluorocarbons'.

This has been fixed.

Figures: All of the figures need a lot of work in my opinion. The panels are small and difficult to read. For the draft they could have used more of the width of the paper. There is also quite a lot of white space in between some panels. There is a lack of tick marks (and even axes) on some plots (e.g. Figure 3).

Figures have been remade following the reviewer's suggestion. Tick labels are omitted when redundant to remove white space.

Line 116. This note about the two isomers of CFC-113 should be made and explained earlier.

This has been moved to where CFC-113 is first introduced (Line 55).

Figure 6 and estimate of equivalent ODP emissions. Panel B shows that the major contributor is CFC-115 due to the larger emissions in HFC-125 production and the ODP value of 0.45. However, CFC-115 has a lifetime of 540 years, which is much longer than CFC-113 etc, and also very long compared to the timescale for expected recovery of the

ozone layer. The CFC-115 emitted now will in effect give a very slow (small) but long-term depletion of ozone (although a question is how ODP values will differ over such a long timescale). I realise that this is complex and raises a number of issues but I don't think it makes sense to calculate a simple ODP-weighted metric without some comment and maybe also present the results without CFC-115.

This is a good point. We have added the following at the beginning of Section 2.4 to address this (Lines 288–292):

“Note that the lifetimes of CFC-113, CFC-114, and CFC-115 are longer than that of CFC-11, which ODP is calculated in reference to, so these numbers do not reflect the timing of ozone depletion. In particular, CFC-115 decays with a lifetime of 540 years, meaning that its impact on ozone depletion will be small in the near-term and largely after the expected return to 1980 stratospheric chlorine and bromine levels.”

Reviewer #2 (Remarks to the Author):

Comments on “Bayesian modeling of HFC production pipeline suggests growth in unreported CFC by-product and feedstock production” by Bourguet and Lickley

Evaluation of trends in emissions of controlled substances by the Montreal Protocol with a view to identifying currently unknown emission sources is important, topic and of broad interest. The authors developed a Bayesian framework to quantify emissions of CFCs in HFC production, to explain discrepancies between emissions derived from reported information and observed atmospheric CFCs concentrations. They found the use of CFCs as feedstocks in the HFC production accounts for most of the CFCs emissions under report. Thus, they speculate that there is un-reported feedstock production in A5 countries, which may explain recent unexpected atmospheric growth in CFCs concentrations.

We thank Reviewer 2 for their comments and constructive feedback.

Major comments:

The results of this paper stand on the assumptions that the HFC production is accurate and that non-A5 countries produced and consumed CFC-113 and CFC-114 feedstocks in separate processes. However, I have large concern on both assumptions. The authors also acknowledged some of the uncertainties after the conclusions in section 3.1. Therefore, I suggest the authors to be cautious on the speculations that are made in section 3.1. In terms of the second assumption, the authors showed the consistency between the reported values by non-A5 countries and their modeling results as a validation of the assumption, but this is not robust.

We agree that Section 3.1 benefits from being more cautious and we have updated the text in Section 3.1 following these suggestions.

Figure 4B compares the reported production of CFC-113 (by non-A5 countries) with the BPE estimations for non-A5 countries. The two have a general consistency as most of the reported values falls within the 1-sigma interval of the BPE estimations, but the recent reported increase from 2015-2019 of CFC-113 production by non-A5 countries are not capture by the BPE estimations. This is a flaw that undergrade the confidence of the conclusion (the authors made) that recent increase in CFC-113 is largely from A5 countries. The recent increase may be from end-products that we do not consider (CTFE, HFOs), so we expect the BPE estimate of feedstock production to be biased low relative to reported values (which may include production of CFC-113 as a feedstock for CTFE and HFO). We have added the following sentence to address disagreement between reported and posterior values (Lines 205–212):

“Given that reported values for CFC-113 came only from non-A 5 countries from 2008--2019, this suggests that feedstock production in these countries was likely not underreported during this time. As CFC-113 is known to be used as a feedstock in other emerging production processes, the low bias of the BPE posterior may reflect reporting of CFC-113 for use in these processes and accurate reporting of CFC-113 feedstock use across the industry. It is also possible that the assumed chemical conversion rate for the production of CFC-114 from CFC-113 is lower than our assumed value of 98%, which would increase the mass of CFC-113 feedstock required to produce a given mass of HFC-134a (see Supplementary Fig. 1).”

There is a time lag between the reported values and emissions/atmospheric concentrations considering implementations and atmospheric transport. What is the time the authors assumed for the reporting and atmospheric mixing? I suggest the authors to extend their results to the most recent years, e.g. 2023. What about the most recent reported values by non-A5 countries? It could simply be the continuing relatively-high-level production (~ 100 Gg/year) by the non-A5 countries.

As mixing ratios in year $t+1$ are assumed to increase due to emissions in year t , we assume that atmospheric mixing takes one year. We have added this sentence to the Methods section to clarify this point (Lines 475–477):

“As mixing ratios in year $t + 1$ depend on processes in year t , this formulation assumes a 1-y mixing time in the troposphere.”

In addition to the banks model that we use for non-feedstock and by-product production, we assume that emissions occur in the same year as feedstock and by-products are produced. We have clarified this in the following sentence (508–509):

“For simplicity, we assume that all feedstock and by-product emissions occur in the same year that the feedstocks and by-products are produced.”

We chose to not extend mixing ratio simulations beyond 2020 (representing emissions through 2019) for two reasons. First, much of the CFC-113 and CFC-114 data from AGAGE has been flagged in recent years (https://agage2.eas.gatech.edu/data_archive/global_mean/readme). Luke Western stated (through personal communication) that AGAGE data beyond 2021 are not reliable at this time, and so we used only mixing ratios that were previously published up to 2020 in Western et al. (2023). Second, the HFC production data from Velders et al. (2022) is projected from 2020 onwards and would therefore not capture Covid-related anomalies in 2020. Thus, HFC-125 and HFC-134a production is not constrained by observations and/or

reporting during that time, limiting its accuracy as an emission source in our simulations. We have added the following sentence to the Methods section to address this (Lines 469–471):

“Global mixing ratios of CFC-113 and CFC-114 have been published through 2020 but are not publicly available beyond that year. The period of our analysis is therefore 1990–2020.”

Another issue is about the inconsistency between the BPE posteriors with observationally derived emissions of CFCs (Figure 3 right column). I think there is no significant increase in the observational derived emissions, especially for CFC-113 and HCFC-133a, as the interannual variability is large. The BPE posterior clearly did not capture the large interannual variability. The authors claim they “do not impact variability beyond interannual timescales”. Is it true? The unexplained interannual variability adds to the uncertainty in long-term change, leading to a low confidence in the increase trend showed by BPE posterior emissions. It is especially concerning when the studied period is relatively short (16 years from 2004 to 2019).

Emissions plotted in Figs. 1, 3, 4, and 5 are now 5-yr running means (2019 is a 3-yr mean). Although some variability remains, the positive trend in non-bank emissions for each of the CFCs is more pronounced (Fig. 1A).

We condition our posterior on mixing ratios, not emissions, and we have an autocorrelation term that accounts for some of the interannual variability that could arise from atmospheric variability or variability in industrial processes. The BPE model is not designed to fit each year's data, but rather capture the underlying trends in mixing ratios, which we have. However, the reviewer is right that we are not able to explain all of the interannual variability with our model, and we modified the text to clarify this (Lines 157–169):

“Despite this smoothing, some variability remains that is not captured by our simulation posteriors, particularly for CFC-113 and HCFC-133a. It is possible that facility-level drivers of interannual emissions variability, such as leakage during maintenance or improvements to containment following modernization, could be a source of this variability -- the BPE model assumes constant emission rates from all sources, and thus cannot capture these effects. The model also uses HFC production time series that are partly informed by a top-down emission estimate, which would not capture potential temporal misalignment in production and emissions, although we anticipate that 5-y averaging should smooth out this variability. CFC-113 may also have been used to a lesser extent during this time as a feedstock or intermediate in the production of other end-products, such as chlorotrifluoroethylene (CTFE) plastics, trifluoroacetic acid (TFA), and the hydrofluoroolefin HFO-1336mzz(Z).

Thus, the remaining variability in observationally-derived emissions could be from several sources that do not impact the overall trend of emissions associated with HFC production.”

Specific comments:

Lines 12-14: Can you make this clearer? For example, adding something like “increasing” or their positive trend or increase rates here. What are the values of “expected given global reporting”?

This sentence has been modified to (Lines 12–14):

“Recent observations of three chlorofluorocarbons (CFCs), CFC-113, CFC-114, and CFC-115, suggest that emissions of these compounds have not decreased as expected given global reporting of their production.”

Due to disparities in trends across gases, and the abstract word length, we feel that greater detail is not possible in the abstract.

Line 28, Change “When” to “Once”

This substitution would not change the meaning of the sentence, and we prefer our choice of wording to maintain a consistent style of writing.

Figure 1: Can you add some explanation about the possible causes of annual variation of CFC-113 in Figure 1A, while less variability for CFC-114 and CFC-115?

Figure 1A now shows the 5-y running mean of non-bank emissions. CFC-113 still has higher variability than CFC-114 and CFC-115, and this is explored in Section 2.1 (Lines 155–169):

“As the central aim of this work is to capture underlying multi-year trend in emissions, rather than year-to-year variability, the 5-y running mean of observationally-derived emissions is shown. Despite this smoothing, some variability remains that is not captured by our simulation posteriors, particularly for CFC-113 and HCFC-133a. It is possible that facility-level drivers of interannual emissions variability, such as leakage during maintenance or improvements to containment following modernization \cite{vollmer2015}, could be a source of this variability -- the BPE model assumes constant emission rates from all sources, and thus cannot capture these effects. The model also uses HFC production time series that are partly informed by a top-down emission estimate, which would not capture potential temporal misalignment in production and emissions, although we anticipate that 5-y averaging should smooth out this variability. CFC-113 may also

have been used to a lesser extent during this time as a feedstock or intermediate in the production of other end-products, such as chlorotrifluoroethylene (CTFE) plastics, trifluoroacetic acid (TFA), and the hydrofluoroolefin HFO-1336mzz(Z) \cite{TEAP2020,andersen2021,rust2023}. Thus, the remaining variability in observationally-derived emissions could be from several sources that do not impact the overall trend of emissions associated with HFC production.”

Lines 169-177, Can you explain how these values are calculated? They are hardly to be estimated from Figure 3.

Emissions during each time period are summed, and the emissions from the source of interest are divided by the total estimated emissions. This may be difficult to do visually using Figure 3, but it is done using the data that are used to make Figure 3.

Lines 199-202: Note the uncertainty is very large. According to the BPE priors, the CFC-113 and CFC-114 production in A5 countries changes from 9-47% (2004-2008) to 35-80% (2015-2019). Is the “increase” statistically significant?

Wording has been changed to reflect uncertainty in this conclusion (Lines 215–217):

“Following from the previously reported estimate of HFC production in A 5 and non-A 5 countries used to inform our priors, 63% (32–87%) of CFC-113 and CFC-114 production occurred in A 5 countries from 2015–2019, up from 27% (9–59%) in 2004--2008, reflecting a potential increase in the mass of global feedstock production that was not reported.”

Tables 1 and 2, Can you also include the emission range here?

We considered this, however, because emissions is a time series rather than constant parameters, we didn’t think it belonged in either table. However, we note that we’ve shown the emissions ranges in Figs. 4 and 5 for HFC-134a and HFC-125, respectively, and they are stated for HFC-134a in Section 2.2. To minimize repetition, we have left them out of Tables 1 and 2.

I am not sure it is a good idea to estimate the global warming potentials for the future scenarios because there are many assumptions used.

We have removed our estimates of future GWP and ODP following this suggestion.

Figures 1, 3, 5, 6: The end year “2019” should be noted on the time axis.

These figures have been updated accordingly.

Reviewer #2 (Remarks on code availability):

I have downloaded them but have not run the code since it is written in MATLAB.

Reviewer #4 (Remarks to the Author):

This manuscript improves our understanding of the increasing emissions of several CFCs, by examining their emission breakdowns from allowed feedstock use, by-product emissions and consumption/banks etc., following a previously developed Bayesian inference framework. The study throws light on the exempted usage of ODSs under the Montreal Protocol, and by-product emissions of ODSs, which are somehow ignored and could be potential remaining challenges of the ozone layer recovery, and provides useful tool for potential further studies. The authors also acknowledge the limitations and uncertainties within the modeling framework of the study. The manuscript is generally well-structured and well-written. I believe the manuscript is suitable for publication in Nature Communications, after some revisions.

My specific comments are below:

1. I think the authors need to present a comparison of the posterior values with prior values of each variable included in the Bayesian, for a complete picture of the performance of Bayesian. It is unclear now whether some of the conclusions are due to the constraint by the observations, or due to the prior values.

E.g. Lines 203ff. The posterior emission rate seems to be close to the mean of the prior distribution. It is not clear whether the posterior emission rates are due to constraint, or prior. I would suggest adding a comparison of the fit to observations using prior and posterior parameters.

Prior and posterior distributions for input parameters are now included in Figs. S3–5.

Specify what variables were constrained in Methods/paragraph 112ff.

This is now included in Table 3, which describes the input parameters and marks which ones are updated in the BPE simulations.

Tabulate the input parameters and the definitions of all non-feedstock end uses.

Input parameters are now tabulated in Table 3. Non-feedstock end uses are now broadly defined as “short-lived banks and long-lived banks.” Readers can refer to Lickley et al. (2022), which is cited when declaring bank types, for further description of non-feedstock end-uses.

Line 527ff: Why the prior FE is 0-4% lower than the 1.5-6.2%? what is the distribution for prior feedstock emission rate?

FE for CFC-113 is 1.0-6.3%. For CFC-114 and HCFC-133a, the following has been updated in the Methods section for clarity (Lines 548–551):

“For computational efficiency, after simulating each gas independently, the FEk114 and FEk133a parameter spaces were updated to remove the tails of the parameter space where the conditional probability of the data given the parameter value was near zero. As FEk114 and FEk133a posteriors suggested values lower than the MCTOC range, these distributions were also adjusted to include values between 0 and 1.5%.”

It is also noted later in that paragraph (Line 557):

“We assumed beta distributions with parameters (2, 2) for FE.”

Section 2.2-2.3. I suggest the authors also discuss the annual changes in the posterior feedstock/byproduct emission rates.

We assume fixed emission rates in A 5 and non-A 5 countries. The global average rate may change as emissions shift to A 5 countries, but there is not enough industry information to implement annually-variable emission rates. This has been clarified when describing the BPE model in the Methods section (Lines 497–498):

“DE, RF, FE, and BP are all assumed to be constant with time.”

The following sentence has also been added to the captions of Tables 1 and 2 to emphasize this (Line 250 and 262):

“Non-A 5 and A 5 emission rates do not vary with time, while the global emission rates may vary as production shifts from non-A 5 to A 5 countries.”

2. Some of the wording needs some clarification.

E.g. both “feedstock usage”, “feedstock production” and “feedstock” appear in the text. I understand the feedstock production is used in the Bayesian calculation for this process, but I think the authors need to clarify the statement of feedstock production/consumption/usage and where emissions will happen related to this process somewhere in the text.

We cannot determine where emissions occur in the feedstock production/consumption pipeline with this formulation but rather, we quantify their emissions in aggregate. We have added the following sentence to the Introduction to clarify terminology (Lines 68–70):

“Given that emissions can occur at any point during the production, distribution, and consumption of feedstocks, we refer to all emissions from this pipeline feedstock emissions.”

Additionally, “feedstock production” is now used specifically when discussing the mass of feedstocks produced, and “production and consumption” is only used when distinguishing between those processes. This language has been updated throughout the text.

E.g. Line 174 “HFC-125 production” -> “byproduct emissions from HFC-125 production”. Same for 176, 177 and elsewhere in the text. Always specify the emission information in the text to make things clear.

E.g. Line 241: “CFC-113 and CFC-114 emission rates” -> “byproduct emission rates during HFC-125 production”

These examples are fixed and wording is updated throughout the text.

3. The last sentence of abstract: “underscores the importance of the HFC production phasedowns...”. The major contribution from phasing down HFC production is HFC itself (from HFC consumption), thus not a finding of this study. I suggest reconsidering this statement.

This sentence has been modified to (Lines 24–27):

“Nonetheless, this work demonstrates the environmental impacts of tightened ODS feedstock regulations and adds a reduction in CFC emissions to the benefits of the HFC production phasedowns scheduled by the Kigali Amendment.”

4. Reference 1 - Suggest referring to the specific chapter of the Ozone Assessment report to credit the chapter authors, as done in ref 23, 7, 64.

Reference 1 has been updated.

5. Figure 1 caption: The authors use “portion” here and elsewhere several times. I think using portion (usually for percentage) to represent the absolute emissions may lead to some ambiguity.

“Portion of” has been removed here and replaced or removed when necessary in the text.

6. Line 124: “in their production”, what does “their” refer to?

This has been clarified (“their” refers to HCFC-123 and HCFC-124 production here).

7. Line 126: “feedstock production and by-product emission rates”, of the CFCs? And I think the feedstock production are both constrained?

I assume this was intended to point out that feedstock emission rates are also constrained.

That is now included in this sentence (Line 126–129):

“Previously reported estimates of HFC-134a and HFC-125 production in Article 5 (A 5; low- to middle-income) and non-Article 5 (non-A 5; high-income) countries

are used to jointly model and constrain feedstock production and feedstock and by-product emission rates from the manufacturing pipeline in the two classifications of countries.”

8. Fig 3. You need to explain the legend in the figure caption, such as “no HFC production”. “Production” should be non-feedstock production.

An explanation has been added to the caption of Fig. 3:

“In the left column, No HFC production refers to mixing ratio simulations that use posterior emissions from non-HFC sources only, while the emissions for these simulations are shown by the Non-HFC production/banks curves in the right column.”

9. The uncertainties ranges in the text and tables: the authors should explain what the uncertainties are, especially for the average values over a time period. The description in Table 1 looks a bit confusing.

Table captions have been updated for clarity.

10. Line 228: why using “observationally-derived” emissions here not the BPE estimated emissions?

This was an error – values were relative to BPE estimated emissions. This has now been corrected in the text.

11. Section 2.4 and corresponding Methods part: I believe that ODP and GWP terms are a fix term for a substance representing their capacity of depleting the ozone layer and radiative forcing. I do not think the use of “ODP of emissions” is appropriate. It is confusing especially you use “ODP” for both actually ODP and ODP-eq emissions in the text. The ODP of emissions -> The ODP-weighted emissions or ODP-Gg emissions. ODP has been changed to “ODP-weighted emissions” and GWP has been changed to “CO2-equivalent” where necessary.

Also, define “unintended” here.

This is now defined with the parenthetical: “(i.e., feedstock and by-product)”

12. Lines 350ff: however, your conclusion from the posterior emission rate in the above section (lines 203ff) suggest that: the posterior feedstock emission rate is low and not supporting the transport emissions between production and consumption. Does it mean that your emission rate results support that CFCs are used as intermediate and not subject to reporting both in A5 and non-A5 countries?

We assume that non-A 5 countries would not have reported feedstock production if CFCs were used as intermediates. The following sentence has been added to address the possibility of CFC-114 being used as an intermediate in both A 5 and non-A 5 countries (365–367):

“As CFC-114 estimated emission rates are at the low end of what is thought to be technically feasible, it is possible that some amount of CFC-114 feedstock production was produced and consumed as an intermediate.”

13. Eq(6), should be FE133a. Missing “a”.

“a” has been added.

14. Line 554-556. What is the “autocorrelation term” here?

The autocorrelation term reflects the correlated error between modeled and observed mixing ratios.

15. Line 585-586. What is the 0.6-0.8 here and how it compares to the 0.95 in Line 571?

Have the authors considered a year-decay correlation coefficient?

The 0.6 - 0.8 range refers to the autocorrelation of the error between modeled and observed mixing ratios for HCFC-133a. This term has a lower autocorrelation than that of CFCs (which we set to 0.95). We have added some text to make this clarification (Lines 607–610):

“This measurement uncertainty, level of temporal aggregation, and potential interannual variability that is not captured by our model also cause the off-diagonal autocorrelation term within Sigma_133a to be uncertain and lower than that of the CFCs. Therefore, we modeled this autocorrelation term as a beta distribution between 0.6--0.8 with parameters (2, 2).”

Reviewer #4 (Remarks on code availability):

The information of the Code is for "PNAS submission". Not sure if this should be corrected.

This has been corrected.

Reviewer #2 (Remarks to the Author):

Comments on “Bayesian modeling of HFC production pipeline suggests growth in reported CFC by-product and feedstock production” by Bourguet and Lickley

Thank the authors for having addressed some initial concerns and comments in the revised version. Some important clarifications have been added. However, there are still a few major points outlined below which required to be addressed to ensure the article is ready for publication.

We thank Reviewer 2 for another thoughtful review. We have addressed their comments and believe the manuscript has improved as a result.

1. Figure 1 appears improved by smoothing, but it primarily reveals a high level of uncertainty relative to a minor increase, particularly for CFC-113. This suggests that the observed increase is likely not significant. The authors claim, ‘CFC-113, CFC-114, and CFC-115 emissions ... increased between 2004 and 2019.’ However, is this increase truly significant? Given the high uncertainty and limited sample size, any confident conclusions regarding long-term changes are questionable and may reflect a fundamental flaw in the analysis.

We thank the reviewer for bringing this point to our attention, but we disagree that the uncertainty is a fundamental flaw in our analysis. The relevant statistical test would be to determine whether the trend in reported production is contained within the possible trends in observationally-derived non-bank emissions. We describe below why we do not feel this is necessary.

Reported production of CFC-113 and CFC-114 decreased from 2004–2019 (Fig. 1B), and no CFC-115 production was reported during this time. Based on our assumption of a constant bank emission rate, bank emissions of these compounds also decreased over this time due to the decreasing bank size. Therefore, if reporting were accurate, we would expect observationally-derived emissions to decrease due to both decreasing feedstock production and decreasing bank emissions – but this is not the case. In Fig. 1A, we present non-bank emissions to make a more direct comparison with reported production. It is clear that non-bank emissions did not decrease as reported production did. This disagreement is the motivation for our work, and we do not feel that an evaluation of the trend in emissions strengthens this motivation.

It is also worth noting that the uncertainty that is displayed in Fig. 1A is highly correlated through time. The uncertain parameters in the observationally-derived emissions estimate are constant for each emission inversion, meaning that all years in a given inversion will have the

same bias. In other words, the uncertainty in Fig. 1A reflects an uncertainty in emission magnitude, not an uncertainty in variability.

Nonetheless, we have changed the wording of the sentences in question to highlight the disagreement between estimated non-bank emissions and reported feedstock production (Lines 59–65):

“Previous work has suggested that emissions from banked reservoirs of CFC-113, CFC-114, and CFC-115 cannot explain observationally-derived values (Lickley et al., 2022), and while there was no sign of a decrease in the emissions of these compounds from sources other than banks from 2004–2019 (Fig. 1A), the globally-aggregated feedstock production of CFC-113 and CFC-114 reported to the Ozone Secretariat decreased during this time (Fig. 1B). Thus, an unknown source of emissions may have prevented emissions from decreasing as one would expect based on reporting from 2004–2019.”

In the Line 84, it mentions that "to 500 Gg/year in 2019 shown in Fig.1C-D", this number is the sum from Fig1C and Fig1D?

That is correct. We have clarified this by adding the word “combined”:

“In particular, the estimated growth of combined production of the refrigerants HFC-134a and HFC-125 from around 200 Gg/y in 2004 to 500 Gg/y in 2019 (shown in Fig. 1C–D; data from Velders et al. (2022)) has been associated with the concurrent rise in a suite of CFC emissions.”

2. Similarly, for Figure 3 and any references to “trend” elsewhere in the text, The authors still need to conduct a statistical test to substantiate these trends.

As we discuss above, for each of the references to a “trend,” we do not feel that a statistical test is necessary. The references to “trend” in the text all refer to the discrepancy between observationally-derived emissions and reported feedstock production (Lines 132–136, 154–158, and 351–353). As reported feedstock production clearly decreased from 2004–2019, the relevant statistical test would be to determine whether the trend in reported production is contained within the possible trends in observationally-derived emissions. (A zero trend in observationally-derived emissions would still motivate our analysis.) This does not require proving the statistical significance of the positive trend in observationally-derived emissions, and we feel that it is sufficiently clear that reported feedstock production cannot explain observationally-derived emissions without further quantification.

Lines 132–136:

“By explicitly modeling the conversion and by-production of these CFCs and HCFC-133a through the HFC-125 and HFC-134a manufacturing pipelines in A 5 and non-A 5 countries, we attempt to explain the apparent discrepancy between trends in reported feedstock production and observationally-derived emissions and quantify feedstock and by-product emission rates in each country classification.”

Lines 154–158:

“Relative to previous work (Lickley et al., 2022), the magnitudes and trends of the BPE posterior emission distributions for these gases provide an improved comparison with observationally-derived emissions from 2004–2019 (Fig 3, right column). As the central aim of this work is to capture the underlying multi-year trend in emissions, rather than year-to-year variability, the 5-y running mean of observationally-derived emissions is shown.”

Note: Regardless of the statistical significance of the trend in observationally-derived emissions, our posteriors are better able to reproduce the long-term behavior of the time series relative to Lickley et al. (2022). In that work, which does not consider emissions from HFC production, the posterior emission estimates decrease from 2004–2019, which disagrees with the non-decreasing trend in observationally-derived emissions (see their Fig. 2). Both our emission posterior and the observationally-derived emissions exhibit zero or increasing trends. We feel that this qualitative agreement does not need to be quantified for our conclusion to hold.

Lines 351–353:

“Thus, emissions from unreported feedstock production in A 5 countries may explain the discrepancy between trends in reported feedstock production and observationally-derived emissions during this time.”

3. Line 223 (in track-change version): “Notably, reported CFC-113 production values are greater than our 1-sigma interval for estimated non-A5 production from 2004-2012 and at the high end of the interval from 2013-2019. Given that reported values for CFC-113 came only from non-A5 countries from 2008-2019, this suggests that feedstock production in these countries was likely not underreported during this time” The statement is reasonable. However, it could also imply that the estimated CFC-113 production values might underestimate actual production in non-A5 countries, thereby mistakenly overestimating the contribution from A5 countries.

This is a good point, and something we considered, though it was inconsistent with the assumptions around HFC-134a production. Assuming that our HFC-134a production estimates in non-A5 and A5 countries are accurate, a reduction in CFC-113 production in A5 countries would require a larger portion of HFC-134a production in these countries to occur by the TCE pathway (which emits HCFC-133a). This possibility is included in our prior parameter space, but it is inconsistent with observations of HCFC-133a and therefore has a near zero probability in the posterior. Additionally, reported CFC-113 feedstock production would not be able to explain observationally-derived emissions, so some emissions from non-reported production is needed. Given that HFC-134a was likely produced by the PCE pathway (which emits CFC-113) in A5 countries, it is reasonable to attribute the non-reported feedstock production and its associated emissions to HFC-134a production in A5 countries. Thus, the HFC-134a production from A5 countries is likely not an overestimate.

However, to acknowledge our result's dependence on our HFC-134a production estimate, we have modified the sentence in question to the following:

“Notably, reported CFC-113 production values are greater than our 1-sigma interval for estimated non-A 5 production from 2004–2012 and at the high end of the interval from 2013-2019. Assuming that our HFC-134a production estimates in A 5 and non-A 5 countries are accurate, and given that reported values for CFC-113 came only from non-A5 countries from 2008-2019, this suggests that feedstock production in non-A 5 countries was likely not underreported during this time.”

4. The authors need to double-check the Global production of CFC-113 and CFC-114 in Fig4B and Figure 4C, it seems by eyes that Fig4B of CFC-113 was wrongly plotted which is not consistent with the line Lin 196 (to 205 Gg/y in 2019”).

We thank the reviewer for bringing this to our attention. An updated version of Fig. 4 is now included in the manuscript.

Reviewer #2 (Remarks on code availability):

They can be downloaded but have not tested since I am not a matlab user